# The open XYZ spin 1/2 chain: Separation of variables and scalar products for boundary fields related by a constraint

Giuliano Niccoli[1⋆] and Veronique Terras[2†]

**1** Univ Lyon, Ens de Lyon, Univ Claude Bernard, CNRS,
Laboratoire de Physique, F-69342 Lyon, France
**2** Université Paris-Saclay, CNRS, LPTMS, 91405, Orsay, France

⋆ giuliano.niccoli@ens-lyon.fr , † veronique.terras@universite-paris-saclay.fr

## Abstract

We consider the open XYZ spin chain with boundary fields. We solve the model by the new Separation of Variables approach introduced in [1]. In this framework, the transfer matrix eigenstates are obtained as a particular sub-class of the class of so-called separate states. We consider the problem of computing scalar products of such separate states. As usual, they can be represented as determinants with rows labelled by the inhomogeneity parameters of the model. We notably focus on the special case in which the boundary parameters parametrising the two boundary fields satisfy one constraint, hence enabling for the description of part of the transfer matrix spectrum and eigenstates in terms of some elliptic polynomial $Q$-solution of a usual $TQ$-equation. In this case, we show how to transform the aforementioned determinant for the scalar product into some more convenient form for the consideration of the homogeneous and thermodynamic limits: as in the open XXX or XXZ cases, our result can be expressed as some generalisation of the so-called Slavnov determinant.



# 1   Introduction

In this work, we solve the XYZ open spin chain with arbitrary boundary fields by the new Separation of Variables approach introduced in [1,2] and compute the scalar product of separate states. Our aim is to generalise the results obtained in the XXX and XXZ cases [3–5] to the open XYZ case, i.e. to obtain convenient scalar product representations for the consideration of the homogeneous and thermodynamic limits.

For the study of quantum integrable models, and in particular for the explicit computation of physical quantities such as correlation functions, it is especially important to be able to compute scalar products of some class of states which includes the eigenstates of the model. For simple models solved by algebraic Bethe Ansatz (ABA), such as the XXZ spin 1/2 chain or the Lieb-Liniger model with periodic boundary conditions, the determinant representation initially obtained by N. Slavnov [6] for the scalar products of Bethe states, one of which being on-shell, has turned to be one of the key-ingredients enabling the computation of form factors and correlation functions [7–33]. Its generalisation to more complicated models which, in the (generalised) ABA context, is usually far from being obvious, has therefore been at the centre of an important activity recently [34–45]. In particular, the obtention of such a formula for the XYZ periodic model has remained for decades a particularly challenging problem due to the combinatorial complexity of the construction of the Bethe states in this model [46,47], so that progresses in this respect could have been made only very recently [42,45]. Concerning the XYZ chain with open boundary conditions, an ABA solution was proposed in [48] for boundary fields satisfying some constraint, but the computation of the scalar products of Bethe states within this framework still remains an open problem.

For models solved by Separation of Variables (SoV) [1,2,49–83], the natural class of states generalising the eigenstates are the so-called separate states, which admit factorised wave-functions. It happens that, by construction, the scalar products of such separate states can be expressed as simple determinants, at least for models with a rank one underlying algebra [66,68–74,76,80] (see also [84–86] for higher rank generalisations). However, the form of these determinants is a priori very different from its original ABA counterpart [6] and seems less adapted to the computation of physical quantities such as correlation functions. In fact, it usually depends in an intricate way on the (unphysical) inhomogeneity parameters of the model, which makes it difficult to consider the homogeneous and thermodynamic limits.

The problem of a reformulation of these determinants in a more suitable form for the homogeneous and thermodynamic limits has already been considered for XXX and XXZ spin chains with various types of boundary conditions [3–5,87]. In the later works, a connection

with Slavnov-type determinants has been observed for scalar products of certain classes of separate states, which opened the way to the computation of the model correlation functions within the SoV framework [88–92]. In particular, the scalar product formulas obtained in [5] for the open XXZ chain with non-diagonal boundary conditions enabled us recently to explicitly compute, in the half-infinite chain limit, a class of density matrix elements for some non-parallel boundary fields [91, 92]. In [92] we notably focussed on the interesting case for which the (in general non-longitudinal) boundary fields are related by a constraint so that the spectrum can still be partially described by usual Bethe equations [93–95]. In the present work, we consider the open XYZ chain with boundary fields, our aim being to generalise the results of [5] to this model, so as to open the way to the computation of the correlation functions as in [92]. As in [92], we shall pay a special attention to the case for which the boundary fields are related by a constraint, enabling one to partially describe the spectrum in terms of usual $TQ$ and Bethe equations [48, 96].

The content of the article is the following. In section 2, we introduce the algebraic aspects of the model. In section 3, we present its solution by Separation of Variables. We recall that Sklyanin's version of the SoV approach for the open XYZ spin chain was developed in [73], based on the use of a vertex-IRF transformation [46]. Here, we choose instead to present the solution of the model via a generalisation to the open XYZ chain of the new SoV approach proposed in [1,2] for the XXX and XXZ models, as an alternative to the approach of [73]. The use of the vertex-IRF transformation is not necessary within this new approach, which makes the formal solution presented in section 3 much lighter than the one of [73]. In section 4, we present our main result, concerning determinant representations for the scalar products of separate states. We focus here on the case in which the boundary parameters are related by a constraint (meaning that there are only 5 independent boundary parameters, the last one being fixed in terms of the others by the constraint), and on the scalar products of separate states which can be described by an elliptic polynomial $Q$-function compatible with this constraint. In that case, we can reformulate the scalar products obtained by SoV as some generalised Slavnov determinants. The details of the computations leading to this result are then presented in section 5. It is worth mentioning here that the method we use in this elliptic case is not a direct generalisation of the one used for the open XXX and XXZ chains [4,5]. Instead, it is freely inspired from the one used in [87], however leading here to much simpler results.

## 2 The reflection algebra for the XYZ open spin chain

Let us first introduce some notations.

For a given imaginary parameter $\omega$ ($\Im\omega > 0$), the functions $\theta_j(\lambda|k\omega)$, $j = 1, 2, 3, 4$, $k = 1, 2$, denote the usual theta functions [97] with quasi-periods $\pi$ and $k\pi\omega$. In the following, we shall use shortcut notations for the theta functions with simple and double imaginary quasi-period, and write $\theta_j(\lambda) \equiv \theta_j(\lambda|\omega)$, $\vartheta_j(\lambda) \equiv \theta_j(\lambda|2\omega)$. We shall also simply write $\theta(\lambda) \equiv \theta_1(\lambda)$, and $\theta_s(\lambda, \mu) = \theta(\lambda - \mu)\theta(\lambda + \mu)$.

We introduce the eight-vertex R-matrix $R(\lambda) \in \text{End}(\mathbb{C}^2 \otimes \mathbb{C}^2)$,

$$R(\lambda) = \begin{pmatrix} a(\lambda) & 0 & 0 & d(\lambda) \\ 0 & b(\lambda) & c(\lambda) & 0 \\ 0 & c(\lambda) & b(\lambda) & 0 \\ d(\lambda) & 0 & 0 & a(\lambda) \end{pmatrix}, \tag{1}$$

which corresponds to the elliptic solution of the Yang-Baxter equation (on $\mathbb{C}^2 \otimes \mathbb{C}^2 \otimes \mathbb{C}^2$):

$$R_{12}(\lambda_1 - \lambda_2)R_{13}(\lambda_1 - \lambda_3)R_{23}(\lambda_2 - \lambda_3) = R_{23}(\lambda_2 - \lambda_3)R_{13}(\lambda_1 - \lambda_3)R_{12}(\lambda_1 - \lambda_2). \tag{2}$$

Here a($\lambda$), b($\lambda$), c($\lambda$), d($\lambda$) are given as the following functions of the spectral parameter $\lambda$:

$$a(\lambda) = \frac{2\,\vartheta_4(\eta)\,\vartheta_1(\lambda+\eta)\,\vartheta_4(\lambda)}{\theta_2(0)\,\vartheta_4(0)}, \qquad b(\lambda) = \frac{2\,\vartheta_4(\eta)\,\vartheta_1(\lambda)\,\vartheta_4(\lambda+\eta)}{\theta_2(0)\,\vartheta_4(0)}, \qquad (3)$$

$$c(\lambda) = \frac{2\,\vartheta_1(\eta)\,\vartheta_4(\lambda)\,\vartheta_4(\lambda+\eta)}{\theta_2(0)\,\vartheta_4(0)}, \qquad d(\lambda) = \frac{2\,\vartheta_1(\eta)\,\vartheta_1(\lambda+\eta)\,\vartheta_1(\lambda)}{\theta_2(0)\,\vartheta_4(0)}. \qquad (4)$$

They also depend, in addition to the spectral parameter $\lambda$ and to the elliptic quasi-period $\omega$, on a parameter $\eta \in \mathbb{C}$ which corresponds to the crossing parameter of the model. We shall suppose here that this parameter $\eta$ is generic, or at least that $\mathbb{Z}\eta \cap (\mathbb{Z}\pi + \mathbb{Z}\pi\omega) = \emptyset$. In (2) and in the following, the indices denote as usual on which space(s) of the tensor product the corresponding operator acts. The R-matrix (1) also satisfies the following unitary and crossing symmetry relations on $\mathbb{C}^2 \otimes \mathbb{C}^2$:

$$R_{21}(-\lambda)R_{12}(\lambda) = -\theta_s(\lambda, \eta)\,\mathbb{I}_{12}, \qquad (5)$$

$$R_{12}(\lambda)\,\sigma_1^y\left[R_{12}(\lambda-\eta)\right]^{t_1}\sigma_1^y = \theta_s(\lambda, \eta)\,\mathbb{I}_{12}, \qquad (6)$$

as well as the identities

$$R_{12}(\lambda+\pi) = -\sigma_1^z R_{12}(\lambda)\sigma_1^z, \qquad R_{12}(\lambda+\pi\omega) = -e^{-i(2\lambda+\pi\omega+\eta)}\sigma_1^x R_{12}(\lambda)\sigma_1^x. \qquad (7)$$

We also introduce the following boundary K-matrix,

$$K(\lambda) \equiv K(\lambda; \alpha_1, \alpha_2, \alpha_3) = \frac{\theta_1(2\lambda)}{2\,\theta_1(\lambda)}\left[\mathbb{I} + c^x \frac{\theta_1(\lambda)}{\theta_4(\lambda)}\sigma^x + ic^y \frac{\theta_1(\lambda)}{\theta_3(\lambda)}\sigma^y + c^z \frac{\theta_1(\lambda)}{\theta_2(\lambda)}\sigma^z\right], \qquad (8)$$

with coefficients $c^x, c^y, c^z$ given in terms of three boundary parameters $\alpha_1, \alpha_2, \alpha_3$ as

$$c^x = \prod_{\ell=1}^{3}\frac{\theta_4(\alpha_\ell)}{\theta_1(\alpha_\ell)}, \qquad c^y = -\prod_{\ell=1}^{3}\frac{\theta_3(\alpha_\ell)}{\theta_1(\alpha_\ell)}, \qquad c^z = \prod_{\ell=1}^{3}\frac{\theta_2(\alpha_\ell)}{\theta_1(\alpha_\ell)}. \qquad (9)$$

As found in [98, 99] (we use here for convenience a parametrization close to the one of [96]), this K-matrix corresponds to the scalar solution of the reflection equation [100] with the R-matrix (1):

$$R_{21}(\lambda-\mu)K_1(\lambda)R_{12}(\lambda+\mu)K_2(\mu) = K_2(\mu)R_{21}(\lambda+\mu)K_1(\lambda)R_{12}(\lambda-\mu). \qquad (10)$$

It also satisfies the properties

$$K(\lambda)K(-\lambda) = \prod_{\ell=1}^{3}\frac{\theta_s(\alpha_\ell, \lambda)}{\theta^2(\alpha_\ell)}\,\mathbb{I}, \qquad (11)$$

$$K(\lambda+\pi) = -\sigma^z K(\lambda)\sigma^z, \qquad K(\lambda+\pi\omega) = (-e^{-2i\lambda-i\pi\omega})^3 \sigma^x K(\lambda)\sigma^x. \qquad (12)$$

To take into account the boundary conditions at both ends of the spin chain, we in fact introduce two boundary matrices $K_\pm$, each of them being associated with three boundary parameters $\alpha_1^\pm, \alpha_2^\pm, \alpha_3^\pm$, defining coefficients $c_\pm^x, c_\pm^y, c_\pm^z$ as in (9). They are defined as

$$K_-(\lambda) = K(\lambda-\eta/2; \alpha_1^-, \alpha_2^-, \alpha_3^-), \qquad K_+(\lambda) = K(\lambda+\eta/2; \alpha_1^+, \alpha_2^+, \alpha_3^+). \qquad (13)$$

From the R-matrix (1) and the boundary matrix $K_-$ one can construct, following the method initially proposed in [101], a boundary monodromy matrix $\mathcal{U}_-(\lambda)$ as

$$\mathcal{U}_-(\lambda) = T(\lambda)K_-(\lambda)\hat{T}(\lambda), \qquad (14)$$

in which $T(\lambda)$ is the bulk monodromy matrix defined as

$$T(\lambda) \equiv T_0(\lambda) = R_{01}(\lambda - \xi_1 - \eta/2)\ldots R_{0N}(\lambda - \xi_N - \eta/2), \tag{15}$$

and

$$\hat{T}(\lambda) \equiv \hat{T}_0(\lambda) = (-1)^N \sigma_0^y \, T_0^{t_0}(-\lambda) \, \sigma_0^y = R_{0N}(\lambda + \xi_N - \eta/2)\ldots R_{01}(\lambda + \xi_1 - \eta/2). \tag{16}$$

In these definitions, $\xi_1, \ldots, \xi_N$ are arbitrary inhomogeneity parameters. The boundary monodromy matrix $\mathcal{U}_-(\lambda)$, as well as the bulk monodromy matrix $T(\lambda)$, act on the tensor product $\mathcal{H}_0 \otimes \mathcal{H}$, where $\mathcal{H}_0 \simeq \mathbb{C}^2$ is the so-called auxiliary space, whereas $\mathcal{H} = \otimes_{n=1}^N \mathcal{H}_n$, with $\mathcal{H}_n \simeq \mathbb{C}^2$, is the quantum space of the model. In other words, $\mathcal{U}_-(\lambda)$ and $T(\lambda)$ are to be understood as $2 \times 2$ matrices (as written in the auxiliary space) with operators entries in $\mathcal{H}$, and the products in (14), (15) and (16) are to be understood as matrix multiplication in $\mathcal{H}_0$. In particular, we shall denote

$$\mathcal{U}_-(\lambda) = \begin{pmatrix} \mathcal{A}_-(\lambda) & \mathcal{B}_-(\lambda) \\ \mathcal{C}_-(\lambda) & \mathcal{D}_-(\lambda) \end{pmatrix}. \tag{17}$$

The boundary monodromy matrix $\mathcal{U}_-(\lambda)$ satisfies the reflection equation,

$$R_{21}(\lambda - \mu)\mathcal{U}_{-,1}(\lambda)R_{12}(\lambda + \mu - \eta)\mathcal{U}_{-,2}(\mu) = \mathcal{U}_{-,2}(\mu)R_{21}(\lambda + \mu - \eta)\mathcal{U}_{-,1}(\lambda)R_{12}(\lambda - \mu), \tag{18}$$

as well as the inversion relation

$$\mathcal{U}_-^{-1}(\lambda + \eta/2) = \frac{\theta(2\lambda - 2\eta)}{\det_q \mathcal{U}_-(\lambda)} \mathcal{U}_-(-\lambda + \eta/2) = \frac{\widetilde{\mathcal{U}}_-(\lambda - \eta/2)}{\det_q \mathcal{U}_-(\lambda)}. \tag{19}$$

Here, $\widetilde{\mathcal{U}}_-$ denotes the 'algebraic adjunct' of $\mathcal{U}_-$, defined by

$$\widetilde{\mathcal{U}}_-(\lambda) = \begin{pmatrix} -\mathrm{c}(2\lambda)\mathcal{A}_-(\lambda) + \mathrm{b}(2\lambda)\mathcal{D}_-(\lambda) & -\mathrm{a}(2\lambda)\mathcal{B}_-(\lambda) + \mathrm{d}(2\lambda)\mathcal{C}_-(\lambda) \\ -\mathrm{a}(2\lambda)\mathcal{C}_-(\lambda) + \mathrm{d}(2\lambda)\mathcal{B}_-(\lambda) & -\mathrm{c}(2\lambda)\mathcal{D}_-(\lambda) + \mathrm{b}(2\lambda)\mathcal{A}_-(\lambda) \end{pmatrix}, \tag{20}$$

and $\det_q \mathcal{U}_-$ the quantum determinant, which is a central element of the reflection algebra, and which is given by

$$\det_q \mathcal{U}_-(\lambda) = \det_q T(\lambda) \det_q T(-\lambda) \det_q K_-(\lambda), \tag{21}$$

where

$$\det_q T(\lambda) = a(\lambda + \eta/2)\,d(\lambda - \eta/2), \tag{22}$$

$$a(\lambda) = \prod_{n=1}^N \theta(\lambda - \xi_n + \eta/2), \qquad d(\lambda) = a(\lambda - \eta) = \prod_{n=1}^N \theta(\lambda - \xi_n - \eta/2), \tag{23}$$

and

$$\det_q K_\pm(\lambda) = \theta(2\eta \pm 2\lambda)\prod_{\ell=1}^3 \frac{\theta_s(\lambda, \alpha_\ell^\pm)}{\theta^2(\alpha_\ell^\pm)}. \tag{24}$$

Still following [101], let us finally introduce the transfer matrix

$$\mathcal{T}(\lambda) = \mathrm{tr}\big[K_+(\lambda)\,T(\lambda)K_-(\lambda)\,\hat{T}(\lambda)\big] = \mathrm{tr}\big[K_+(\lambda)\mathcal{U}_-(\lambda)\big], \tag{25}$$

which defines a one-parameter family of commuting operators on $\mathcal{H}$:

$$[\mathcal{T}(\lambda), \mathcal{T}(\mu)] = 0, \quad \forall \lambda, \mu. \tag{26}$$

The Hamiltonian of the open XYZ spin 1/2 chain, which can be obtained in terms of a logarithmic derivative of the transfer matrix (25) in the homogeneous limit ($\xi_n = 0$, $n = 1, \ldots, N$) and at $\lambda = \frac{\eta}{2}$, is given by

$$H = \sum_{a \in \{x,y,z\}} \left[ \sum_{n=1}^{N} J_a \sigma_n^a \sigma_{n+1}^a + h_+^a \sigma_1^a + h_-^a \sigma_N^a \right], \tag{27}$$

in which the coupling constants $J_{x,y,z}$ and the components $h_\pm^{x,y,z}$ of the boundary fields are parameterised as[1]

$$J_x = \frac{\theta_4(\eta)}{\theta_4(0)}, \qquad h_\pm^x = c_\pm^x \frac{\theta_1(\eta)}{\theta_4(0)} = \frac{\theta_1(\eta)}{\theta_4(0)} \prod_{\ell=1}^{3} \frac{\theta_4(\alpha_\ell^\pm)}{\theta_1(\alpha_\ell^\pm)}, \tag{28}$$

$$J_y = \frac{\theta_3(\eta)}{\theta_3(0)}, \qquad h_\pm^y = i c_\pm^y \frac{\theta_1(\eta)}{\theta_3(0)} = -i \frac{\theta_1(\eta)}{\theta_3(0)} \prod_{\ell=1}^{3} \frac{\theta_3(\alpha_\ell^\pm)}{\theta_1(\alpha_\ell^\pm)}, \tag{29}$$

$$J_z = \frac{\theta_2(\eta)}{\theta_2(0)}, \qquad h_\pm^z = c_\pm^z \frac{\theta_1(\eta)}{\theta_2(0)} = \frac{\theta_1(\eta)}{\theta_2(0)} \prod_{\ell=1}^{3} \frac{\theta_2(\alpha_\ell^\pm)}{\theta_1(\alpha_\ell^\pm)}. \tag{30}$$

## 3 Diagonalisation of the transfer matrix by SoV

In this section, we briefly explain how the new SoV approach, proposed in [1] in the open XXX and XXZ cases, can be adapted to characterize the transfer matrix spectrum and eigenstates for the inhomogeneous deformation of the open XYZ model considered here.

### 3.1 Transfer matrix properties

Let us start by recalling some elementary properties of the transfer matrix, which can be easily deduced from the properties of the R-matrix and of the boundary K-matrices. For convenience, we introduce the following notations:

$$\xi_n^{(h)} = \xi_n + \frac{\eta}{2} - h\eta, \qquad h \in \{0,1\}, \quad n \in \{1,\ldots,N\}, \tag{31}$$

$$\xi_0^{(0)} = \frac{\eta}{2}, \qquad \xi_{-1}^{(0)} = \frac{\eta}{2} + \frac{\pi}{2}, \qquad \xi_{-2}^{(0)} = \frac{\eta}{2} + \frac{\pi\omega}{2}, \qquad \xi_{-3}^{(0)} = \frac{\eta}{2} + \frac{\pi + \pi\omega}{2}. \tag{32}$$

**Proposition 3.1.** *The transfer matrix $\mathcal{T}(\lambda)$ is an even elliptic polynomial[2] in $\lambda$ of order $2N + 6$, i.e. it satisfies the following parity and quasi-periodicity properties in $\lambda$ w.r.t. the periods $\pi$ and $\pi\omega$:*

$$\mathcal{T}(-\lambda) = \mathcal{T}(\lambda), \qquad \mathcal{T}(\lambda + \pi) = \mathcal{T}(\lambda), \qquad \mathcal{T}(\lambda + \pi\omega) = \left(-e^{-2i\lambda - i\pi\omega}\right)^{2N+6} \mathcal{T}(\lambda). \tag{35}$$

---

[1] We have exchanged $h_+$ and $h_-$ with respect to standard notations, instead of exchanging '+' and '-' in the K-matrices as done in our previous open XXZ papers [91] and [92].

[2] We say that a function $f$ is an even elliptic polynomial of order (or degree) $2n$ if it is an even function which in addition satisfies the quasi-periodicity properties:

$$f(\lambda + \pi) = f(\lambda), \qquad f(\lambda + \pi\omega) = \left(-e^{-2i\lambda - i\pi\omega}\right)^{2n} f(\lambda). \tag{33}$$

It means that there exist constants $C, \lambda_1, \ldots, \lambda_n$ such that

$$f(\lambda) = C \prod_{j=1}^{n} \theta_s(\lambda, \lambda_j). \tag{34}$$

*Moreover, it satisfies the following quantum determinant property*

$$\mathcal{T}(\xi_n + \eta/2)\,\mathcal{T}(\xi_n - \eta/2) = \frac{\det_q K_+(\xi_n)\,\det_q \mathcal{U}_-(\xi_n)}{\theta(\eta + 2\xi_n)\,\theta(\eta - 2\xi_n)}, \tag{36}$$

*and its value at the special points* (32) *is central and can be evaluated as*

$$\mathcal{T}(\xi_0^{(0)}) = \tau_0\,\mathbb{I}, \quad \text{with } \tau_0 = (-1)^N \frac{\theta(2\eta)}{\theta(\eta)}\,\det_q T(0), \tag{37}$$

$$\mathcal{T}(\xi_{-1}^{(0)}) = \tau_{-1}\,\mathbb{I}, \quad \text{with } \tau_{-1} = c_-^z\,c_+^z\,\frac{\theta(2\eta)}{\theta(\eta)}\,\det_q T\Big(\frac{\pi}{2}\Big), \tag{38}$$

$$\mathcal{T}(\xi_{-2}^{(0)}) = \tau_{-2}\,\mathbb{I}, \quad \text{with } \tau_{-2} = -e^{i\left(2\sum_{n=1}^N \xi_n - (N+3)\eta - \frac{3\pi\omega}{2}\right)} c_-^x\,c_+^x\,\frac{\theta(2\eta)}{\theta(\eta)}\,\det_q T\Big(-\frac{\pi\omega}{2}\Big), \tag{39}$$

$$\mathcal{T}(\xi_{-3}^{(0)}) = \tau_{-3}\,\mathbb{I}, \quad \text{with } \tau_{-3} = e^{i\left(2\sum_{n=1}^N \xi_n - (N+3)\eta - \frac{3\pi\omega}{2}\right)} c_-^y\,c_+^y\,\frac{\theta(2\eta)}{\theta(\eta)}\,\det_q T\Big(-\frac{\pi + \pi\omega}{2}\Big). \tag{40}$$

*Then, under the condition*

$$\epsilon_n \xi_n^{(0)}, \ -3 \le n \le N, \ \epsilon_n \in \{-1, 1\}, \quad \text{are pairwise distinct modulo } (\pi, \pi\omega), \tag{41}$$

*we can write the following interpolation formula for the transfer matrix,*

$$\mathcal{T}(\lambda) = \mathsf{T}_0(\lambda)\,\mathbb{I} + \sum_{n=1}^N r_n(\lambda)\,\mathcal{T}(\xi_n^{(0)}), \tag{42}$$

*in which we have defined the functions*

$$r_n(\lambda) \equiv r_n(\lambda|\xi_1, \ldots, \xi_N) = \prod_{\substack{k=-3 \\ k \neq n}}^{N} \frac{\theta_s(\lambda, \xi_k^{(0)})}{\theta_s(\xi_n^{(0)}, \xi_k^{(0)})}, \qquad -3 \le n \le N, \tag{43}$$

$$\mathsf{T}_0(\lambda) = \sum_{n=0}^{3} r_{-n}(\lambda)\,\tau_{-n}. \tag{44}$$

## 3.2 SoV basis

Let us suppose that the inhomogeneity parameters are generic, or at least that they satisfy the condition

$$\epsilon_n \xi_n^{(h_n)}, \ 1 \le n \le N, \ h_n \in \{0, 1\}, \ \epsilon_n \in \{-1, 1\}, \quad \text{are pairwise distinct modulo } (\pi, \pi\omega), \tag{45}$$

and that the boundary matrices $K_+(\lambda)$ and $K_-(\lambda)$ are not both proportional to the identity. Let **A** be any function of the spectral parameter $\lambda$ satisfying the relation

$$\mathbf{A}(\lambda + \eta/2)\,\mathbf{A}(-\lambda + \eta/2) = \frac{\det_q K_+(\lambda)\,\det_q \mathcal{U}_-(\lambda)}{\theta(\eta + 2\lambda)\,\theta(\eta - 2\lambda)}. \tag{46}$$

Then, one can show similarly as in [1] that, for almost any choice[3] of the co-vector $\langle S|$ and of the inhomogeneity parameters satisfying (45), the states

$$\langle \mathbf{h}| \equiv \langle S| \prod_{n=1}^{N} \left(\frac{\mathcal{T}(\xi_n - \frac{\eta}{2})}{\mathbf{A}(\frac{\eta}{2} - \xi_n)}\right)^{1-h_n}, \qquad \mathbf{h} \equiv (h_1, \ldots, h_N) \in \{0, 1\}^N, \tag{47}$$

---

[3]In [1], the determinant of the matrix formed by the coordinates of the covectors (3.17) in the natural basis is computed and proven nonzero for some special choice of the inhomogeneities and the coordinates of the convector $\langle S|$. Being this determinant a nonzero multivariable polynomial in these parameters, it can be zero only on the codimension 1 hypersurface of its zeros. This means that these covectors form a basis out of this zero hypersurface, that is for almost any value of the inhomogeneities and choice of $\langle S|$.

form a basis of $\mathcal{H}^*$. Let $|R\rangle \in \mathcal{H}$ be the unique vector such that

$$\langle \mathbf{h} | R \rangle = \delta_{\mathbf{h},\mathbf{0}} \, \frac{\mathcal{N}_0}{V(\xi_1^{(0)}, \ldots, \xi_N^{(0)}) \, V(\xi_1, \ldots, \xi_N)}, \tag{48}$$

for some given normalisation coefficient $\mathcal{N}_0$. Here we have used the shortcut notation (31) for the shifted inhomogeneity parameters, and have defined, for any set of variables $\zeta_1, \ldots, \zeta_N$, the quantity,

$$V(\zeta_1, \ldots, \zeta_N) = \prod_{1 \le i < j \le N} \theta_s(\zeta_j, \zeta_i). \tag{49}$$

Then the states

$$|\mathbf{h}\rangle = \prod_{n=1}^{N} \left( \frac{\theta(2\xi_n - 2\eta)}{\theta(2\xi_n + 2\eta)} \frac{\mathcal{T}(\xi_n + \frac{\eta}{2})}{\mathbf{A}(\frac{\eta}{2} - \xi_n)} \right)^{h_n} |R\rangle, \qquad \mathbf{h} \equiv (h_1, \ldots, h_N) \in \{0,1\}^N, \tag{50}$$

form a basis of $\mathcal{H}$, which satisfies with (47) the following orthogonality condition:

$$\langle \mathbf{h} | \mathbf{k} \rangle = \delta_{\mathbf{h},\mathbf{k}} \, \frac{\mathcal{N}_0}{V(\xi_1^{(h_1)}, \ldots, \xi_N^{(h_N)}) \, V(\xi_1, \ldots, \xi_N)}, \qquad \forall \, \mathbf{h}, \mathbf{k} \in \{0,1\}^N. \tag{51}$$

## 3.3  SoV characterisation of the transfer matrix spectrum and eigenstates

The discrete SoV characterisation of the transfer matrix spectrum and eigenstates follows from standard arguments, similar to those developed for the open XXX and XXZ chains in [1], see also [73] for a formulation for the open XYZ chain within Sklyanin's SoV approach.

**Theorem 3.1.** *Let us suppose that the inhomogeneity parameters are generic, or at least that they satisfy the non-intersecting conditions*

$$\begin{aligned} &\epsilon_n \, \xi_n^{(h_n)}, \; 1 \le n \le N, \; h_n \in \{0,1\}, \; \epsilon_n \in \{-1,1\}, \\ \text{and} \quad &\epsilon_j \, \xi_{-j}^{(0)}, \; 1 \le j \le 3, \; \epsilon_j \in \{-1,1\}, \end{aligned} \qquad \text{are pairwise distinct modulo } (\pi, \pi\omega). \tag{52}$$

*Let us also suppose that the boundary matrices $K_+(\lambda)$ and $K_-(\lambda)$ are not both proportional to the identity. Then the transfer matrix $\mathcal{T}(\lambda)$ is diagonalisable with simple spectrum, and its spectrum $\Sigma_{\mathcal{T}}$ is given by the set of functions of the form*

$$\tau(\lambda) = \mathsf{T}_0(\lambda) + \sum_{a=1}^{N} r_n(\lambda) \, \tau_n, \tag{53}$$

*in terms of the functions (43) and (44), where $\tau_1, \ldots, \tau_N$ satisfy the following inhomogeneous system of $N$ quadratic equations:*

$$\tau_n \left( \mathsf{T}_0(\xi_n - \eta) + \sum_{k=1}^{N} r_k(\xi_n - \eta) \, \tau_k \right) = \frac{\det_q K_+(\xi_n) \, \det_q \mathcal{U}_-(\xi_n)}{\theta(\eta + 2\xi_n) \, \theta(\eta - 2\xi_n)}, \quad \forall n \in \{1, \ldots, N\}. \tag{54}$$

*Then, the vectors*

$$\sum_{\mathbf{h} \in \{0,1\}^N} \prod_{n=1}^{N} Q(\xi_n^{(h_n)}) \, V(\xi_1^{(h_1)}, \ldots, \xi_N^{(h_N)}) \, |\mathbf{h}\rangle, \tag{55}$$

$$\sum_{\mathbf{h} \in \{0,1\}^N} \prod_{n=1}^{N} \left[ \left( \frac{\theta(2\xi_n - 2\eta) \, \mathbf{A}(\frac{\eta}{2} + \xi_n)}{\theta(2\xi_n + 2\eta) \, \mathbf{A}(\frac{\eta}{2} - \xi_n)} \right)^{h_n} Q(\xi_n^{(h_n)}) \right] V(\xi_1^{(h_1)}, \ldots, \xi_N^{(h_N)}) \, \langle \mathbf{h} |, \tag{56}$$

with coefficients $Q(\xi_n^{(0)}), Q(\xi_n^{(1)})$ satisfying

$$\frac{Q(\xi_n^{(1)})}{Q(\xi_n^{(0)})} = \frac{\tau(\xi_n + \frac{\eta}{2})}{\mathbf{A}(\frac{\eta}{2} + \xi_n)} = \frac{\mathbf{A}(\frac{\eta}{2} - \xi_n)}{\tau(\xi_n - \frac{\eta}{2})}, \qquad \forall n \in \{1, \dots, N\}, \tag{57}$$

in terms of the function (46), are respectively the right and left $\mathcal{T}$-eigenstate with eigenvalue $\tau \in \Sigma_{\mathcal{T}}$, which are uniquely defined up to an overall normalisation factor.

Let us decompose the function $\mathbf{A}$ (46) as

$$\mathbf{A}(\lambda) = (-1)^N \frac{\theta(2\lambda + \eta)}{\theta(2\lambda)} \mathbf{A}_K(\lambda) \, a(\lambda) \, d(-\lambda), \tag{58}$$

in which

$$\mathbf{A}_K(\lambda + \eta/2) \, \mathbf{A}_K(-\lambda + \eta/2) = \frac{\det_q K_+(\lambda) \det_q K_-(\lambda)}{\theta(2\eta + 2\lambda) \, \theta(2\eta - 2\lambda)}. \tag{59}$$

Then one can easily show that

$$\prod_{n=1}^{N} \left( -\frac{\theta(2\xi_n - \eta)}{\theta(2\xi_n + \eta)} \frac{a(\xi_n + \frac{\eta}{2}) \, d(-\xi_n - \frac{\eta}{2})}{a(-\xi_n + \frac{\eta}{2}) \, d(\xi_n - \frac{\eta}{2})} \right)^{h_n} V(\xi_1^{(h_1)}, \dots, \xi_N^{(h_N)})$$
$$= \frac{V(\xi_1^{(0)}, \dots, \xi_N^{(0)})}{V(\xi_1^{(1)}, \dots, \xi_N^{(1)})} V(\xi_1^{(1-h_1)}, \dots, \xi_N^{(1-h_N)}). \tag{60}$$

Hence the left eigenvector (56) can equivalently be rewritten as

$$\frac{V(\xi_1^{(0)}, \dots, \xi_N^{(0)})}{V(\xi_1^{(1)}, \dots, \xi_N^{(1)})} \sum_{\mathbf{h} \in \{0,1\}^N} \prod_{n=1}^{N} \left[ \left( -\frac{\mathbf{A}_K(\frac{\eta}{2} + \xi_n)}{\mathbf{A}_K(\frac{\eta}{2} - \xi_n)} \right)^{h_n} Q(\xi_n^{(h_n)}) \right] V(\xi_1^{(1-h_1)}, \dots, \xi_N^{(1-h_N)}) \langle \mathbf{h} |. \tag{61}$$

## 3.4 Characterisation in terms of a functional $TQ$-equation

For some choice of $\boldsymbol{\varepsilon} \equiv (\epsilon_1^+, \epsilon_2^+, \epsilon_3^+, \epsilon_1^-, \epsilon_2^-, \epsilon_3^-) \in \{+1, -1\}^6$, let us consider the following simple solution of (59),

$$\mathbf{A}_K(\lambda) \equiv \mathbf{A}_{\boldsymbol{\varepsilon}}(\lambda) = \prod_{\sigma = \pm} \prod_{i=1}^{3} \frac{\theta(\alpha_i^\sigma + \epsilon_i^\sigma(\frac{\eta}{2} - \lambda))}{\theta(\alpha_i^\sigma)} = \prod_{\sigma = \pm} \prod_{i=1}^{3} \frac{\theta(\lambda - \frac{\eta}{2} - \epsilon_i^\sigma \alpha_i^\sigma)}{\theta(-\epsilon_i^\sigma \alpha_i^\sigma)}, \tag{62}$$

and denote

$$\mathbf{A}_{\boldsymbol{\varepsilon}}(\lambda) = (-1)^N \frac{\theta(2\lambda + \eta)}{\theta(2\lambda)} \mathbf{A}_{\boldsymbol{\varepsilon}}(\lambda) \, a(\lambda) \, d(-\lambda). \tag{63}$$

The states (47) and (50) with the choice (63) of the function (46) will be denoted respectively as

$$_{\boldsymbol{\varepsilon}} \langle \mathbf{h} | \qquad \text{and} \qquad | \mathbf{h} \rangle_{\boldsymbol{\varepsilon}}. \tag{64}$$

**Proposition 3.2.** *Under the hypothesis of Theorem 3.1, let us suppose that the six boundary parameters $\alpha_i^\sigma$, $i = 1, 2, 3$, $\sigma = \pm$, satisfy the following constraint:*

$$\sum_{\sigma = \pm} \sum_{i=1}^{3} \epsilon_i^\sigma \alpha_i^\sigma = (N - 2M - 1)\eta, \tag{65}$$

*for some choice of $\boldsymbol{\varepsilon} \equiv (\epsilon_1^+, \epsilon_2^+, \epsilon_3^+, \epsilon_1^-, \epsilon_2^-, \epsilon_3^-) \in \{\pm 1\}^6$ such that $\prod_{\sigma = \pm} \prod_{i=1}^{3} \epsilon_i^\sigma = 1$. Then, any entire function $\tau$ for which there exists a function $Q$ of the form*

$$Q(\lambda) = \prod_{n=1}^{M} \theta_s(\lambda, q_n), \qquad \mathbf{q} = (q_1, \dots, q_M) \in \mathbb{C}^M, \tag{66}$$

*such that $\tau$ and $Q$ satisfy the TQ-equation*

$$\tau(\lambda)Q(\lambda) = \mathbf{A}_{\boldsymbol{\varepsilon}}(\lambda)Q(\lambda-\eta) + \mathbf{A}_{\boldsymbol{\varepsilon}}(-\lambda)Q(\lambda+\eta), \qquad (67)$$

*is an eigenvalue of the transfer matrix (i.e $\tau \in \Sigma_{\mathcal{T}}$), which can therefore be written in the form (53). Moreover, in that case, the function $Q$ (66) satisfying (67) with $\tau$ is unique, and the corresponding right and left eigenvectors of $\mathcal{T}$ are given by (55) and (56).*

*Proof.* The proof of this proposition is standard. We have mainly to observe that, if an entire function $\tau(\lambda)$ satisfies the *TQ*-equation (67) under the condition (65) and with $Q$ of the form (66), then it is an even elliptic polynomial in $\lambda$ of order $2N + 6$. Moreover, the identities

$$\tau_{-i} = \mathbf{A}_{\boldsymbol{\varepsilon}}(\xi_{-i}^{(0)}) \frac{Q(\xi_{-i}^{(0)} - \eta)}{Q(\xi_{-i}^{(0)})}, \qquad \mathbf{A}_{\boldsymbol{\varepsilon}}(-\xi_{-i}^{(0)}) = 0, \qquad \forall i \in \{0, 1, 2, 3\}, \qquad (68)$$

which can be verified by direct computations, imply that $\tau(\lambda)$ has the form (53). Hence we can apply our previous theorem and observe that the identities (57) are verified as consequence of the *TQ*-equation computed in the points $\pm\xi_n^{(0)}$ and $\pm\xi_n^{(1)}$ for any $n \in \{1, \ldots, n\}$. $\qquad\square$

Except in the exceptional case of coinciding roots, the corresponding Bethe equations for the roots $q_j$ of $Q$ (66) solution of (67) under the constraint (65) are therefore

$$\mathbf{A}_{\boldsymbol{\varepsilon}}(q_j)Q(q_j-\eta) + \mathbf{A}_{\boldsymbol{\varepsilon}}(-q_j)Q(q_j+\eta) = 0, \qquad 1 \le j \le M. \qquad (69)$$

The constraint (65) is the direct XYZ analog of Nepomechie's constraint [93, 94] for the XXZ open spin chain. Such kind of constraint already appeared in different contexts in the literature: for instance within the ABA study of the eight-vertex model [48], or within the direct study of the *TQ*-equation [96].

Such a description of the transfer matrix spectrum and eigenstates, in terms of even elliptic polynomial $Q$-functions solving a homogeneous *TQ*-equation (67) in a given sector $M < N$, is of course not complete. A complete description of the general open XYZ transfer matrix spectrum in terms of $Q$-function solutions of a homogeneous[4] *TQ*-equation is still unknown. Note however that, if the constraint (65) is satisfied with some choice of $M$ and $\boldsymbol{\varepsilon}$, it is also satisfied, with the same boundary parameters, for $M' = N-1-M$ and $\boldsymbol{\varepsilon}' = -\boldsymbol{\varepsilon}$. On the basis of a numerical analysis, it was conjectured in [94,95] in the XXZ case and in [96] in the XYZ case that the solutions $Q(\lambda)$ (66) of degree $2M$ of (67) with some particular choice of $\boldsymbol{\varepsilon}$, together with the solutions of degree $2M'$ with $M' = N-1-M$ and $\boldsymbol{\varepsilon}' = -\boldsymbol{\varepsilon}$, provide the complete spectrum and eigenstates of the transfer matrix.

## 4 Scalar products of separate states: Main results

In this section, we present the main results of our paper, namely the determinant representations that we have obtained for the scalar product of separate states.

As usual for models considered in the SoV framework, the scalar products of separate states can be represented as determinants with rows labelled by the inhomogeneity parameters

---

[4]It is also possible to have, in general XYZ open spin chains, a formulation of the spectrum in terms of elliptic polynomial solutions of a *TQ*-equation with an additional inhomogeneous term, as proposed in [102] (see also [103] where another inhomogeneous formulation was derived - and proved to be complete - from SoV in the general open XXZ case). Besides the presence of the inhomogeneous term, the *TQ*-equation of [102] also differs from (66)-(67) by the fact that it involves a couple of non-even related $Q$-functions, instead of a single even one. We do not consider this kind of formulation in our present paper.

(see Proposition 4.1). For physical applications such as the computations of form factors and correlations functions in models for which the eigenstates can be characterised by solutions of Bethe equations, it turns out to be convenient to use instead determinant representations with rows and columns labelled by the two sets of Bethe roots, the prototype of which being the celebrated Slavnov determinant representation [6]. Focusing on cases for which eigenstates of the model can be described by usual Bethe equations, namely under the constraint (65), we present in Theorem 4.1 the analog of such a Slavnov determinant representation for the scalar product of two separate states, one of which being an eigenstate of the model (see also Theorem 4.2 for the orthogonality of the two different sectors which can be described by the same constraint). As in [8], the scalar product is expressed in terms of a Jacobian of the transfer matrix eigenvalues. This generalises to the open XYZ case the formulas obtained in [4,5] in the open XXX and XXZ cases, which played a fundamental role in the computation of the correlation functions in [91,92]. It is however worth mentioning that the approach we used to derive this result from (78) is *not* the direct analog of the approach used in the XXX and XXZ cases. This new approach will be detailed in the section 5.

Let $_\varepsilon\langle P|$ and $|Q\rangle_{\varepsilon'}$ be two arbitrary separate states of the form

$$|Q\rangle_{\varepsilon'} = \sum_{\mathbf{h}\in\{0,1\}^N} \prod_{n=1}^N Q(\xi_n^{(h_n)}) V(\xi_1^{(h_1)},\ldots,\xi_N^{(h_N)}) |\mathbf{h}\rangle_{\varepsilon'}, \tag{70}$$

$$_\varepsilon\langle P| = \frac{1}{\mathcal{N}_0} \sum_{\mathbf{h}\in\{0,1\}^N} \prod_{n=1}^N \left[ \left( -\frac{A_\varepsilon(\frac{\eta}{2}+\xi_n)}{A_\varepsilon(\frac{\eta}{2}-\xi_n)} \right)^{h_n} P(\xi_n^{(h_n)}) \right] V(\xi_1^{(1-h_1)},\ldots,\xi_N^{(1-h_N)}) \,_\varepsilon\langle \mathbf{h}|, \tag{71}$$

for some $\varepsilon,\varepsilon' \in \{\pm 1\}^6$. Here $A_\varepsilon$ denotes the solution of (59) given by (62), whereas $_\varepsilon\langle \mathbf{h}|$ and $|\mathbf{h}\rangle_{\varepsilon'}$, $\mathbf{h} \equiv (h_1,\ldots,h_N) \in \{0,1\}^N$, stand for the states (47) and (50) with function $A$ (46) given by $A_\varepsilon$ and $A_{\varepsilon'}$ respectively.[5] Note that we have

$$|\mathbf{h}\rangle_{\varepsilon'} = \prod_{n=1}^N \left( \frac{A_\varepsilon(\frac{\eta}{2}-\xi_n)}{A_{\varepsilon'}(\frac{\eta}{2}-\xi_n)} \right)^{h_n} |\mathbf{h}\rangle_\varepsilon, \tag{72}$$

so that the separate state (70) can alternatively be written on the basis given by $|\mathbf{h}\rangle_\varepsilon$, $\mathbf{h} \in \{0,1\}^N$, as

$$|Q\rangle_{\varepsilon'} = \sum_{\mathbf{h}\in\{0,1\}^N} \prod_{n=1}^N \left[ \left( \frac{A_\varepsilon(\frac{\eta}{2}-\xi_n)}{A_{\varepsilon'}(\frac{\eta}{2}-\xi_n)} \right)^{h_n} Q(\xi_n^{(h_n)}) \right] V(\xi_1^{(h_1)},\ldots,\xi_N^{(h_N)}) |\mathbf{h}\rangle_\varepsilon. \tag{73}$$

We recall that, up to the choice of a global normalisation, the eigenstates of the transfer matrix are particular cases of the above separate states, for $P$ and $Q$ satisfying (57) with the functions $A_\varepsilon$ and $A_{\varepsilon'}$ respectively. Our aim is therefore to compute the scalar product $_\varepsilon\langle P|Q\rangle_{\varepsilon'}$ of $_\varepsilon\langle P|$ and $|Q\rangle_{\varepsilon'}$.

## 4.1 The original SoV determinant representation

It follows from (51) that the scalar product of (71) and (73) is given by

$$_\varepsilon\langle P|Q\rangle_{\varepsilon'} = \,_\varepsilon\langle Q|P\rangle_{\varepsilon'} \tag{74}$$

$$= \prod_{n=1}^N P(\xi_n^{(0)}) Q(\xi_n^{(0)}) \sum_{\mathbf{h}\in\{0,1\}^N} \prod_{n=1}^N \left[ -\frac{A_\varepsilon(\frac{\eta}{2}+\xi_n)}{A_{\varepsilon'}(\frac{\eta}{2}-\xi_n)} \frac{P(\xi_n^{(1)})Q(\xi_n^{(1)})}{P(\xi_n^{(0)})Q(\xi_n^{(0)})} \right]^{h_n} \frac{V(\xi_1^{(1-h_1)},\ldots,\xi_N^{(1-h_N)})}{V(\xi_1,\ldots,\xi_N)}.$$

---

[5]More generally, the states $_\varepsilon\langle \mathbf{h}|$ and $|\mathbf{h}\rangle_\varepsilon$ in (70) and (73) may stand for any SoV basis elements diagonalising the transfer matrix $\mathcal{T}(\lambda)$ as in Theorem 3.1, provided that it satisfies (51) for a given normalisation constant $\mathcal{N}_0$, and that it also satisfies (72). For instance, such a basis could also be constructed as in [73] by using the Vertex-IRF transformation.

As usual for models solved by SoV, we can rewrite this expression in the form of a simple determinant, which depends however in a intricate way on the inhomogeneity parameters. In the present case, it is a consequence of the following lemma, which can be proven by standard arguments, similarly as in Proposition 4 of [104]:

**Lemma 4.1.** *Let $\Xi_N$ be the space of even elliptic polynomials of order $2(N-1)$, i.e. of holomorphic functions $f$ on $\mathbb{C}$ satisfying the properties*

$$f(-u) = f(u), \qquad f(u+\pi) = f(u), \qquad f(u+\pi\omega) = (-e^{-2iu-i\pi\omega})^{2(N-1)}f(u), \qquad (75)$$

*and let $\{\Theta_N^{(j)}\}_{1\leq j\leq N}$ be a basis of $\Xi_N$. Then, there exists a constant $C$ such that, for any set of complex variables $x_1,\ldots,x_N$,*

$$V(x_1,\ldots,x_N) = C \det_{1\leq i,j\leq N}\left[\Theta_N^{(j)}(x_i)\right]. \qquad (76)$$

It follows from Lemma 4.1 that, for any function $F$ and any set of complex variables $x_1,\ldots,x_N$,

$$\sum_{\mathbf{h}\in\{0,1\}^N}\prod_{n=1}^{N}F(x_n)^{h_n}\frac{V(x_1^{(1-h_1)},\ldots,x_N^{(1-h_N)})}{V(x_1,\ldots,x_N)} = \frac{\det_{1\leq i,j\leq N}\left[\Theta_N^{(j)}(x_i-\frac{\eta}{2})+F(x_i)\Theta_N^{(j)}(x_i+\frac{\eta}{2})\right]}{\det_{1\leq i,j\leq N}\left[\Theta_N^{(j)}(x_i)\right]}, \qquad (77)$$

in which we have used the shortcut notation $x_i^{(h)} = x_i + \frac{\eta}{2} - h\eta$, $1\leq i\leq N$, $h\in\{0,1\}$. Here $\{\Theta_N^{(j)}\}_{1\leq j\leq N}$ stands for any basis of $\Xi_N$, and the ratio (77) is independent of the choice of the basis $\{\Theta_N^{(j)}\}_{1\leq j\leq N}$ of $\Xi_N$. Applying (77) to (74), we therefore obtain the following determinant representation for the scalar product:

**Proposition 4.1.** *Let $_{\boldsymbol{\varepsilon}}\langle P|$ and $|Q\rangle_{\boldsymbol{\varepsilon}'}$ be two arbitrary separate states of the form (71) and (70) for some $\boldsymbol{\varepsilon},\boldsymbol{\varepsilon}'\in\{\pm1\}^6$. Then the scalar product (74) can be rewritten as*

$$_{\boldsymbol{\varepsilon}}\langle P|Q\rangle_{\boldsymbol{\varepsilon}'} = \frac{\det_{1\leq i,j\leq N}\left[P(\xi_i^{(0)})Q(\xi_i^{(0)})\Theta_N^{(j)}(\xi_i^{(1)})-\frac{A_{\boldsymbol{\varepsilon}}(\frac{\eta}{2}+\xi_i)}{A_{\boldsymbol{\varepsilon}'}(\frac{\eta}{2}-\xi_i)}P(\xi_i^{(1)})Q(\xi_i^{(1)})\Theta_N^{(j)}(\xi_i^{(0)})\right]}{\det_{1\leq i,j\leq N}\left[\Theta_N^{(j)}(\xi_i)\right]}, \qquad (78)$$

*in which $\{\Theta_N^{(j)}\}_{1\leq j\leq N}$ stands for any basis of $\Xi_N$, and the ratio (78) is independent of the choice of the basis $\{\Theta_N^{(j)}\}_{1\leq j\leq N}$.*

*Remark* 1. At this stage, we have not made any hypothesis on the particular form of the functions $P$ and $Q$. They can be any arbitrary functions on the discrete set of parameters $\xi_n^{(h_n)}$, $1\leq n\leq N$, $h_n\in\{0,1\}$, such that the two vectors (71) and (70) are well-defined. Also, the boundary parameters are here completely arbitrary.

*Remark* 2. The expression (78) being independent of the choice of the basis $\{\Theta_N^{(j)}\}_{1\leq j\leq N}$ of $\Xi_N$, we are free to choose the most convenient basis according to our purpose. For instance we could choose

$$\Theta_N^{(j)}(u) = \frac{\theta_1^{2(j-1)}(u)\,\theta_4^{2(N-j)}(u)}{\theta_4^{N-1}(0)}, \quad j=1,\ldots,N, \qquad (79)$$

which corresponds to a rewriting of (49) as a generalised Vandermonde determinant:

$$V(\zeta_1,\ldots,\zeta_N) = \prod_{1\leq i<j\leq N}\theta_s(\zeta_j,\zeta_k) = \det_{1\leq i,j\leq N}\left[\frac{\theta_1^{2(j-1)}(\zeta_i)\,\theta_4^{2(N-j)}(\zeta_i)}{\theta_4^{N-1}(0)}\right]. \qquad (80)$$

Another natural choice (that we shall use in Section 5), is to consider a basis of $\Xi_N$ given by functions of the form

$$\Theta^{(j)}_{N,W}(\lambda) = \prod_{\substack{k=1 \\ k \neq j}}^{N} \theta_s(\lambda, w_k) = \frac{W(\lambda)}{\theta_s(\lambda, w_j)}, \quad j = 1, \ldots, N, \qquad \text{with} \quad W(\lambda) = \prod_{j=1}^{N} \theta_s(\lambda, w_j), \quad (81)$$

for some set of parameters $w_1, \ldots, w_N$ such that $\pm w_j$, $1 \leq j \leq N$, are pairwise distinct modulo $\pi$ and $\pi\omega$.

## 4.2 Slavnov's type determinant representation

We now consider functions $P$ and $Q$ of the form

$$P(\lambda) = \prod_{j=1}^{M} \theta_s(\lambda, p_j), \qquad Q(\lambda) = \prod_{j=1}^{M'} \theta_s(\lambda, q_j), \quad (82)$$

for some sets of roots $p_1, \ldots, p_M$ and $q_1, \ldots, q_{M'}$, and we more particularly focus on special cases of interest in which the constraint (65) is satisfied. Our aim is to transform the determinant representation (78) into some more convenient form for the computation of physical quantities such as correlation functions, similarly as what has been done in the open XXX and XXZ cases [4, 5].

More precisely, we focus here on the two following special cases which are compatible with the constraint (65) and the rewriting of eigenstates in terms of functions of the form (82):

1. $\varepsilon = \varepsilon'$, and $M = M'$, with the constraint (65) being satisfied.

2. $\varepsilon' = -\varepsilon$, and $M' + M = N - 1$, so that, in particular, if the constraint (65) is satisfied for $(M, \varepsilon)$, it is still satisfied for $(M', \varepsilon')$.

In the first case, we obtain the following determinant representation for the scalar product (78), which is similar to the representations that have been obtained in the open XXX and XXZ cases:

**Theorem 4.1.** *Let us suppose that the boundary parameters satisfy the constraint (65) for a given $\varepsilon$ and a given $M$, and let $\{q_1, \ldots, q_M\}$ be a solution of the Bethe equations (69), implying that $Q(\lambda)$ of the form (66) is a solution of the homogeneous TQ-equation (67) and that the associated left and right separate states $|Q\rangle_\varepsilon$ and $_\varepsilon\langle Q|$ are transfer matrix eigenstates with eigenvalue $\tau_Q$. Let $\{p_1, \ldots, p_M\}$ be arbitrary parameters, and let $_\varepsilon\langle P|$ be the left separate state of the form (71) associated with the elliptic polynomial (82) of roots $p_1, \ldots, p_M$. Then*

$$\frac{_\varepsilon\langle P|Q\rangle_\varepsilon}{_\varepsilon\langle Q|Q\rangle_\varepsilon} = \frac{_\varepsilon\langle Q|P\rangle_\varepsilon}{_\varepsilon\langle Q|Q\rangle_\varepsilon} = \frac{V(q_1, \ldots, q_M)}{V(p_1, \ldots, p_M)} \prod_{j=1}^{M} \frac{\theta_s(2q_j, \eta)}{\theta_s(2p_j, \eta)} \frac{\det_M\left[\mathcal{S}_Q(\mathbf{p}, \mathbf{q})\right]}{\det_M\left[\mathcal{S}_Q(\mathbf{q}, \mathbf{q})\right]}. \quad (83)$$

*The elements of the $M \times M$ matrix $\mathcal{S}_Q(\mathbf{p}, \mathbf{q})$ are given in terms of the $M$-tuples $\mathbf{p} = (p_1, \ldots, p_M)$ and $\mathbf{q} = (q_1, \ldots, q_M)$ as*

$$\begin{aligned}
\left[\mathcal{S}_Q(\mathbf{p}, \mathbf{q})\right]_{i,j} &= Q(p_i) \frac{\partial \tau_Q(p_i)}{\partial q_j} \\
&= \mathbf{A}_\varepsilon(p_i) Q(p_i - \eta)\left[t(p_i + q_j - \eta/2) - t(p_i - q_j - \eta/2)\right] \\
&\quad - \mathbf{A}_\varepsilon(-p_i) Q(p_i + \eta)\left[t(p_i + q_j + \eta/2) - t(p_i - q_j + \eta/2)\right],
\end{aligned} \quad (84)$$

*in which we have defined*

$$t(\lambda) = \frac{\theta'(\lambda - \frac{\eta}{2})}{\theta(\lambda - \frac{\eta}{2})} - \frac{\theta'(\lambda + \frac{\eta}{2})}{\theta(\lambda + \frac{\eta}{2})}. \tag{85}$$

*In the limit $p_i \to q_i$, $1 \le i \le M$, the elements of the matrix* (84) *become*

$$\left[ \mathcal{S}_Q(\mathbf{q}, \mathbf{q}) \right]_{i,j} = \mathbf{A}_\varepsilon(-q_i) Q(q_i + \eta)$$
$$\times \left[ -\delta_{j,k} \frac{\partial}{\partial \mu} \left( \log \frac{\mathbf{A}_\varepsilon(\mu) Q(\mu - \eta)}{\mathbf{A}_\varepsilon(-\mu) Q(\mu + \eta)} \right)_{\mu = q_i} + K(q_i - q_j) - K(q_i + q_j) \right], \tag{86}$$

*in which we have defined*

$$K(\lambda) = t(\lambda + \eta/2) + t(\lambda - \eta/2) = \frac{\theta'(\lambda - \eta)}{\theta(\lambda - \eta)} - \frac{\theta'(\lambda + \eta)}{\theta(\lambda + \eta)}. \tag{87}$$

*Remark* 3. We have chosen to present here directly a formula for the scalar product $_\varepsilon \langle P | Q \rangle_\varepsilon$ normalised by the "square of the norm"[6] $_\varepsilon \langle Q | Q \rangle_\varepsilon$ since this is the typical kind of ratios that one encounters when computing correlation functions. In other words, this means that the non-renormalised scalar product $_\varepsilon \langle P | Q \rangle_\varepsilon$ is expressed as

$$_\varepsilon \langle P | Q \rangle_\varepsilon = \frac{\mathsf{c}_Q \det_M \left[ \mathcal{S}_Q(\mathbf{p}, \mathbf{q}) \right]}{V(p_1, \ldots, p_M) \prod_{j=1}^M \theta_s(2p_j, \eta)}, \tag{88}$$

with $\mathsf{c}_Q$ being a normalisation coefficient which does not depend on $P$. This coefficient $\mathsf{c}_Q$ is computable, but it has a quite complicated expression (see for instance (123) in the simplest case $M = N/2$), and its knowledge is not necessary for physical applications such as the computation of the correlation functions.

In the second case, we obtain the following orthogonality property of the two different sectors $(M, \varepsilon)$ and $(M', \varepsilon')$ for $\varepsilon' = -\varepsilon$ and $M' = N - M - 1$:

**Theorem 4.2.** *Let $_\varepsilon \langle P |$ and $| Q \rangle_{\varepsilon'}$ be two arbitrary separate states of the form* (71) *and* (70)*, with $P$ and $Q$ being functions of the form* (82)*. We moreover suppose that $\varepsilon' = -\varepsilon$ and $M' + M = N - 1$. Then, the scalar product of $_\varepsilon \langle P |$ and $| Q \rangle_{\varepsilon'}$ vanishes:*

$$_\varepsilon \langle P | Q \rangle_{-\varepsilon} = 0. \tag{89}$$

*Remark* 4. The orthogonality property of Theorem (4.2) is valid under completely general boundary conditions, we do not necessarily suppose here the constraint to be verified. However, an important hypothesis is the form of the functions $P$ and $Q$ as even elliptic polynomials as in (82).

The proof of these results is presented in the next section.

## 5 Scalar products of separate states: Details of the computations

We now explain in this section the different steps of the computations of the scalar products, from the original SoV representation (78) to the representations obtained in Theorem 4.1 and Theorem 4.2.

---

[6] The terminology "square of the norm" is of course abusive since, from our construction of the left/right eigenstates, the quantity $_\varepsilon \langle Q | Q \rangle_\varepsilon$ has no reason to be positive.

## 5.1 Proof of theorem 4.1 in the case $2M = N$

In this section, as well as in the sections 5.2 and 5.3, we consider the case $\boldsymbol{\varepsilon} = \boldsymbol{\varepsilon}'$, $P$ and $Q$ being of the same order $2M$. Since $\boldsymbol{\varepsilon}$ is fixed, we shall slightly simplify the notations and denote $\{a_i\}_{i=1,\dots 6} = \{-\epsilon_i^\sigma \alpha_i^\sigma\}_{i=1,2,3;\sigma=\pm}$, so that

$$\frac{\mathsf{A}_\epsilon(\frac{\eta}{2}+\xi_n)}{\mathsf{A}_{\epsilon'}(\frac{\eta}{2}-\xi_n)} = \frac{\mathsf{A}_\epsilon(\frac{\eta}{2}+\xi_n)}{\mathsf{A}_\epsilon(\frac{\eta}{2}-\xi_n)} = \prod_{\sigma=\pm}\prod_{i=1}^3 \frac{\theta(\xi_n - \epsilon_i^\sigma \alpha_i^\sigma)}{\theta(-\xi_n - \epsilon_i^\sigma \alpha_i^\sigma)} = \prod_{i=1}^6 \frac{\theta(a_i + \xi_n)}{\theta(a_i - \xi_n)}. \tag{90}$$

With these notations, the constraint (65) enabling one to express the spectrum in terms of the homogeneous $TQ$-equation (67) is written as

$$\sum_{\ell=1}^6 a_\ell = (2M + 1 - N)\eta, \tag{91}$$

and the Bethe equations (69) for the roots $q_1, \dots, q_M$ of $Q$ are, for $1 \le i \le M$,

$$\sum_{\sigma=\pm} \sigma \prod_{n=1}^6 \theta(\sigma q_i - \eta/2 + a_n)\, a(\sigma q_i)\, d(-\sigma q_i)\, Q_i(q_i - \sigma\eta) = 0, \tag{92}$$

in which we have used the shortcut notation

$$Q_i(\lambda) = \frac{Q(\lambda)}{\theta_s(\lambda, q_i)}. \tag{93}$$

Let us start in this section by proving Theorem 4.1 in the simple case $2M = N$. Let us first suppose that all the parameters $p_1, \dots, p_M, q_1, \dots, q_M, \xi_1, \dots, \xi_N$ are generic. Then, by choosing a basis of the form (81), with

$$W(\lambda) = P(\lambda)Q(\lambda), \qquad \text{i.e.} \quad (w_1, \dots, w_N) = (p_1, \dots, p_M, q_1, \dots, q_M). \tag{94}$$

in the expression (78) of the scalar product of $_{\boldsymbol{\varepsilon}}\langle P|$ and $|Q\rangle_{\boldsymbol{\varepsilon}}$, we obtain[7]

$$\begin{aligned}
_{\boldsymbol{\varepsilon}}\langle P|Q\rangle_{\boldsymbol{\varepsilon}} &= (-1)^N \prod_{n=1}^N \frac{(PQ)(\xi_n - \frac{\eta}{2})(PQ)(\xi_n + \frac{\eta}{2})}{(PQ)(\xi_n)\prod_{i=1}^6 \theta(\xi_n - a_i)} \frac{\det_N \mathcal{M}}{\det_{1\le i,j\le N}\left[\frac{1}{\theta_s(\xi_i, w_j)}\right]} \\
&= (-1)^N \prod_{n=1}^N \frac{(PQ)(\xi_n - \frac{\eta}{2})(PQ)(\xi_n + \frac{\eta}{2})}{\prod_{i=1}^6 \theta(\xi_n - a_i)} \frac{\det_N \mathcal{M}}{V(w_1, \dots, w_N)V(\xi_N, \dots, \xi_1)},
\end{aligned} \tag{95}$$

in which we have used the explicit expression for the generalised Cauchy determinant

$$\det_{1\le i,j\le N}\left[\frac{1}{\theta_s(\xi_i, w_j)}\right] = \frac{V(w_1, \dots, w_N)V(\xi_N, \dots, \xi_1)}{\prod_{i,j=1}^N \theta_s(\xi_i, w_j)}. \tag{96}$$

In (95), $\mathcal{M}$ is the $N \times N$ matrix with elements

$$\mathcal{M}_{i,j} = \mathsf{M}_{\{a\}}(\xi_i, w_j), \tag{97}$$

in which we have defined the function

$$\mathsf{M}_{\{a\}}(u, v) = \sum_{\epsilon=\pm} \epsilon\, \frac{\prod_{n=1}^6 \theta(u + \epsilon a_n)}{\theta_s(u + \epsilon\frac{\eta}{2}, v)}. \tag{98}$$

---

[7]Here and in the following we are using the compact notation $(PQ)(x) = P(x)Q(x)$.

The determinant of this matrix can be considered as a generalised version of the elliptic Filali determinant [105], dressed here by the product involving the boundary parameters.

Note that similar representations were obtained in our previous works [4, 5]. In these works concerning the XXX and XXZ open spin chains, the strategy to further transform these scalar product representations was then to find an identity enabling one to exchange the role of the two variables in the expression for the matrix elements (the analog of (98)), and therefore of the two sets of variables $\{\xi\}$ and $\{p\} \cup \{q\}$ in the analog of (95) and (78), see equation (D.9) of [5]. The resulting determinant representation for the scalar product was then already explicitly regular with respect to the homogeneous limit, and could be further transformed to a determinant of Slavnov's type, see the details in [4,5]. The difficulty, for the generalisation of this approach to other models, is precisely that the obtention of the key exchange identity is model-dependent, and may therefore not be possible (at least without involving the whole determinant) if the matrix elements are not symmetric enough. In fact, in the present most general elliptic case, for which the expression (97)-(98) explicitly involves 6 boundary parameters (instead of 4, as in the open XXZ case considered in [5]), we could not find a simple exchange identity such as (D.9) of [5]. A direct elliptic analog of the identity (D.9) of [5] can nevertheless be formulated in the case of only 4 boundary parameters related by the constraint, i.e. in the case, under the constraint, in which two of the 6 boundary parameters are opposite of each others and can therefore be simplified from the expression of (97)-(98). In that special case, the strategy of [4,5] can effectively be applied.[8]

To tackle the most general case that we want to consider here, with the 6 boundary parameters being completely arbitrary except for their relation by the constraint (65) (or equivalently (91)), we have therefore developed a different approach. The idea is here to pass quasi-directly from the representation (95) to the Slanov-type determinant by multiplying the matrix $\mathcal{M}$ of elements $\mathsf{M}_{\{a\}}(\xi_i, w_j)$ by a judiciously chosen matrix $\mathcal{X}$. The matrix $\mathcal{X}$ should be chosen so that its columns are labelled by the inhomogeneity parameters, and so that the matrix elements of the product $\mathcal{X} \cdot \mathcal{M}$ can be computed in a simple way by Cauchy's residue theorem, and simplifies when the Bethe equations are satisfied. This approach is freely inspired from the one used in [87] for the antiperiodic XXZ model, the formulas of [87] being however quite different from those of the present case.

---

[8]More precisely, if we suppose, with the notations of (90), that $a_6 = -a_5$, then the quantity $\theta_s(\xi_i, a_5)$ can be factorised out of the expression of the function $\mathsf{M}_{\{a\}}(\xi_i, w_j)$ (97). The 4 remaining parameters satisfy the sum rule $\sum_{\ell=1}^4 a_\ell = \eta$. Under this constraint, one can easily show the following identity, which is the direct elliptic analog of the identity (D.9) of [5]:

$$\frac{1}{\theta(2u)} \sum_{\sigma=\pm} \sigma \frac{\prod_{\ell=1}^4 \theta(u + \sigma a_\ell)}{\theta_s(u + \sigma \frac{\eta}{2}, v)} = \frac{1}{\theta(2v)} \sum_{\sigma=\pm} \sigma \frac{\prod_{\ell=1}^4 \theta(v + \sigma(\frac{\eta}{2} - a_\ell))}{\theta_s(v + \sigma \frac{\eta}{2}, u)}. \tag{99}$$

This identity enables us to directly exchange the role of the parameters $\xi_i$ and $w_j$ in the expression (98) of the matrix $\mathcal{M}$ (97) when the set $\{a\}$ contains only 4 parameters instead of 6, provided we also exchange $\{a\}$ with $\{\eta/2 - a\}$. In other words, it means that the scalar product (78) can alternatively be expressed (up to an easily computable factor) as the following ratio of determinants:

$$_\varepsilon \langle P | Q \rangle_\varepsilon \propto \frac{\det_{1 \le i,j \le 2M}\left[ \Theta_{2M}^{(j)}(w_i - \frac{\eta}{2}) - \prod_{\ell=1}^4 \frac{\theta(w_i + \frac{\eta}{2} - a_\ell)}{\theta(w_i - \frac{\eta}{2} - a_\ell)} \prod_{n=1}^N \frac{\theta_s(w_i - \frac{\eta}{2}, \xi_n)}{\theta_s(w_i + \frac{\eta}{2}, \xi_n)} \Theta_{2M}^{(j)}(w_i + \frac{\eta}{2}) \right]}{\det_{1 \le i,j \le 2M}\left[ \Theta_{2M}^{(j)}(\xi_i) \right]}, \tag{100}$$

which does not depend on the choice of the basis $\Theta_{2M}^{(j)}$. We obtain also similar expressions in cases for which $2M \ne N$ by means of Lemma 5.2 and by using similar arguments as in Appendix D of [5]. Finally, such an expression can be transformed into a Slavnov-type determinant of the form (84) by proceeding as in Appendix E of [5], up to some little subtleties related to the fact that we have to deal here with theta functions instead of hyperbolic ones. However, it seems that the identity (99) cannot easily be generalised to the case of 6 boundary parameters, which explains why we had to proceed differently in the general case.

To this aim, let us introduce, for a generic parameter $\gamma$, the following basis $\{\bar{\Theta}^{(j)}_{2M}\}_{1\le j\le 2M}$ of $\Xi_{2M} \equiv \Xi_N$, defined by

$$\bar{\Theta}^{(j)}_{2M}(\lambda) \equiv \bar{\Theta}^{(j)}_{2M,Q,\gamma}(\lambda) = \frac{Q(\lambda+\frac{\eta}{2})Q(\lambda-\frac{\eta}{2})\,\theta_s(\lambda,\gamma)}{\theta_s(\lambda,z_j)\,\theta_s(\lambda,\tilde{z}_j)}, \qquad 1 \le j \le 2M, \tag{101}$$

with

$$(z_j,\tilde{z}_j) = \begin{cases} (q_j+\frac{\eta}{2}, q_j-\frac{\eta}{2}), & \text{if } j \le M, \\ (q_{j-M}+\frac{\eta}{2},\gamma), & \text{if } j > M, \end{cases} \tag{102}$$

and let us consider the matrix $\mathcal{X} \equiv \mathcal{X}_{Q,\gamma}$ of elements

$$\mathcal{X}_{i,k} \equiv \left[\mathcal{X}_{Q,\gamma}\right]_{i,k} = \frac{\bar{\Theta}^{(i)}_{2M}(\xi_k)}{\theta_s(\xi_k,\gamma)\,\theta(2\xi_k)\prod_{n\ne k}\theta_s(\xi_k,\xi_n)}, \qquad 1 \le i,k \le 2M. \tag{103}$$

The determinant of $\mathcal{X}$ is nonzero and can easily be computed.[9] Note however that its exact value is unimportant for the proof of Theorem 4.1, since it does not depend on $P$ and therefore simplifies in the normalised expression (83), as we shall see later.

The elements of matrix $\mathcal{X} \cdot \mathcal{M}$ are then given by

$$[\mathcal{X} \cdot \mathcal{M}]_{i,j} = \sum_{k=1}^{2M} \mathcal{X}_{i,k} \cdot \mathcal{M}_{k,j} = \sum_{k=1}^{2M} \text{Res}(\mathsf{F}_{i,j};\xi_k), \tag{104}$$

in which we have defined the following functions:

$$\begin{aligned}
\mathsf{F}_{i,j}(\lambda) &\equiv \mathsf{F}_{Q,\{a\},\{\xi\}}(\lambda;z_i,\tilde{z}_i,w_j) \\
&= \frac{\theta'(0)\,Q(\lambda+\frac{\eta}{2})Q(\lambda-\frac{\eta}{2})}{\theta_s(\lambda,z_i)\,\theta_s(\lambda,\tilde{z}_i)\prod_{n=1}^{2M}\theta_s(\lambda,\xi_n)} \sum_{\epsilon=\pm}\epsilon\,\frac{\prod_{n=1}^{6}\theta(\lambda+\epsilon a_n)}{\theta_s(\lambda+\epsilon\frac{\eta}{2},w_j)}.
\end{aligned} \tag{105}$$

Note that, under the constraint

$$\sum_{\ell=1}^{6} a_\ell = \eta, \tag{106}$$

which corresponds to (91) for $N=2M$, and for all $1 \le i,j \le N$, the function $\mathsf{F}_{i,j}$ (105) satisfies the properties

$$\mathsf{F}_{i,j}(-\lambda) = -\mathsf{F}_{i,j}(\lambda), \qquad \mathsf{F}_{i,j}(\lambda+\pi) = \mathsf{F}_{i,j}(\lambda), \qquad \mathsf{F}_{i,j}(\lambda+\pi\omega) = \mathsf{F}_{i,j}(\lambda), \tag{107}$$

so that it is an odd elliptic function of $\lambda$. Hence, the sum of the residues at its poles in an elementary cell vanishes. Moreover, if $\mathsf{F}_{i,j}$ has a pole at some point $x$, it has also a pole at $-x$, and

$$\text{Res}(\mathsf{F}_{i,j};-x) = \text{Res}(\mathsf{F}_{i,j};x). \tag{108}$$

These properties enables one to compute the matrix elements in (104), given by the sum over the residues of $\mathsf{F}_{i,j}$ at $\xi_k$, $1 \le k \le 2M$, in terms of the sums over the residues at the other poles of $\mathsf{F}_{i,j}$. This leads to the following block matrix for the product $\mathcal{X} \cdot \mathcal{M}$:

$$\mathcal{X} \cdot \mathcal{M} = \begin{pmatrix} \mathcal{G}^{(1,1)} & \mathcal{G}^{(1,2)} \\ \mathcal{G}^{(2,1)} & \mathcal{G}^{(2,2)} \end{pmatrix}, \tag{109}$$

---

[9] The determinant $\det_{1\le i,k\le N}[\bar{\Theta}^{(i)}_N(x_k)]$, for arbitrary values of $x_1,\ldots,x_N$, is given up to a constant by Lemma 4.1. The constant can then be fixed by choosing appropriate values for the $x_i$: for $(x_1,\ldots,x_N) = (q_1-\frac{\eta}{2},\ldots,q_M-\frac{\eta}{2},q_1+\frac{\eta}{2},\ldots,q_M+\frac{\eta}{2})$ the matrix becomes triangular and the determinant is given by the product of its diagonal elements. The explicit expression for the determinant of $\mathcal{X}$ follows easily.

with $\mathcal{G}^{(1,1)}$, $\mathcal{G}^{(1,2)}$, $\mathcal{G}^{(2,1)}$ and $\mathcal{G}^{(2,2)}$ being blocks of size $M \times M$. Explicitly, we obtain, for $1 \le i, j \le M$:

$$
\begin{aligned}
\left[\mathcal{G}^{(1,1)}\right]_{i,j} &= \sum_{k=1}^{2M} \text{Res}(\mathsf{F}_{i,j}; \xi_k) = -\text{Res}(\mathsf{F}_{i,j}; p_j + \eta/2) - \text{Res}(\mathsf{F}_{i,j}; p_j - \eta/2) \\
&= \frac{Q_i(p_j)}{\theta(2p_j)} \sum_{\sigma=\pm} \frac{\sigma\, Q_i(p_j + \sigma\eta) \prod_{n=1}^{6} \theta(p_j + \sigma\frac{\eta}{2} - \sigma a_n)}{a(\sigma p_j) d(-\sigma p_j)},
\end{aligned}
\tag{110}
$$

$$
\begin{aligned}
\left[\mathcal{G}^{(1,2)}\right]_{i,j} &= \sum_{k=1}^{2M} \text{Res}(\mathsf{F}_{i,j+M}; \xi_k) = -\delta_{i,j}\left[\text{Res}(\mathsf{F}_{i,j+M}; q_i + \eta/2) + \text{Res}(\mathsf{F}_{i,j+M}; q_i - \eta/2)\right] \\
&= \delta_{i,j} \frac{Q_i(q_i)}{\theta(2q_i)} \sum_{\sigma=\pm} \frac{\sigma\, Q_i(q_i + \sigma\eta) \prod_{n=1}^{6} \theta(q_i + \sigma\frac{\eta}{2} - \sigma a_n)}{a(\sigma q_i) d(-\sigma q_i)},
\end{aligned}
\tag{111}
$$

$$
\begin{aligned}
\left[\mathcal{G}^{(2,1)}\right]_{i,j} &= \sum_{k=1}^{2M} \text{Res}(\mathsf{F}_{i+M,j}; \xi_k) = -\sum_{\sigma=\pm} \text{Res}(\mathsf{F}_{i+M,j}; p_j + \sigma\eta/2) - \text{Res}(\mathsf{F}_{i+M,j}; \gamma) \\
&= \frac{Q(p_j)}{\theta(2p_j)} \sum_{\sigma=\pm} \frac{\sigma\, Q(p_j + \sigma\eta) \prod_{n=1}^{6} \theta(p_j + \sigma\frac{\eta}{2} - \sigma a_n)}{\theta_s(p_j + \sigma\frac{\eta}{2}, q_i + \frac{\eta}{2})\, \theta_s(p_j + \sigma\frac{\eta}{2}, \gamma)\, a(\sigma p_j) d(-\sigma p_j)} \\
&\quad - \mathsf{X}_\gamma(q_i + \eta/2)\, \mathsf{M}_{\{a\}}(\gamma, p_j),
\end{aligned}
\tag{112}
$$

$$
\begin{aligned}
\left[\mathcal{G}^{(2,2)}\right]_{i,j} &= \sum_{k=1}^{2M} \text{Res}(\mathsf{F}_{i+M,j+M}; \xi_k) = -\delta_{i,j}\, \text{Res}(\mathsf{F}_{i+M,j+M}; q_i + \eta/2) - \text{Res}(\mathsf{F}_{i+M,j}; \gamma) \\
&= \delta_{i,j} \frac{Q_i(q_i + \eta)\, Q_i(q_i)}{a(q_i) d(-q_i)} \frac{\theta(\eta) \prod_{n=1}^{6} \theta(q_i + \frac{\eta}{2} - a_n)}{\theta_s(q_i + \frac{\eta}{2}, \gamma)} \\
&\quad - \mathsf{X}_\gamma(q_i + \eta/2)\, \mathsf{M}_{\{a\}}(\gamma, q_j),
\end{aligned}
\tag{113}
$$

in which we have used the shortcut notation

$$
\mathsf{X}_\gamma(\lambda) = \frac{Q(\gamma + \frac{\eta}{2})\, Q(\gamma - \frac{\eta}{2})}{\theta_s(\gamma, \lambda)\, \theta(2\gamma) \prod_{n=1}^{2M} \theta_s(\gamma, \xi_n)}.
\tag{114}
$$

Without any particular assumption on the parameters, i.e. keeping $q_1, \ldots, q_M$ generic, it is possible to further transform the above determinant of size $2M$ by using, as in [3,5], the usual formula for the determinant of block matrices:

$$
\det_{2M}[\mathcal{X} \cdot \mathcal{M}] = \det_M \mathcal{G}^{(2,2)} \cdot \det_M\left[\mathcal{G}^{(1,1)} - \mathcal{G}^{(1,2)}(\mathcal{G}^{(2,2)})^{-1}\mathcal{G}^{(2,1)}\right].
\tag{115}
$$

Since $\mathcal{G}^{(2,2)}$ is the sum of a diagonal invertible matrix and a rank one matrix, it is possible to compute explicitly its determinant by the matrix determinant lemma and its inverse by the Sherman–Morrison formula. This leads to some generalised version of a Slavnov-type determinant, however quite complicated (see [5] in which such kind of generalised formula was presented in the open XXZ case).

This formula simplifies drastically when we particularise $\{q_1, \ldots, q_M\}$ to be a solution of the Bethe equations (92): in that case, the expression (111) vanishes, so that $\mathcal{G}^{(1,2)} = 0$, and therefore

$$
\det_N[\mathcal{X} \cdot \mathcal{M}] = \det_M\left[\mathcal{G}^{(1,1)}\right] \cdot \det_M\left[\mathcal{G}^{(2,2)}\right].
\tag{116}
$$

As mentioned just above, the determinant of the matrix $\mathcal{G}^{(2,2)}$ can explicitly be computed. However, its exact value is unimportant for the computation of the ratio (83), since $\mathcal{G}^{(2,2)}$ does

not depend on $P$. The elements of the matrix $\mathcal{G}^{(1,1)}$ can be rewritten as

$$
\left[\mathcal{G}^{(1,1)}\right]_{i,j} = \frac{-Q(p_j)}{a(p_j)d(-p_j)a(-p_j)d(p_j)} \sum_{\sigma=\pm} \frac{a(\sigma p_j)d(-\sigma p_j)\prod_{n=1}^{6}\theta(\sigma p_j - \frac{\eta}{2} + a_n)Q_i(p_j - \sigma\eta)}{\theta(2\sigma p_j)\,\theta_s(p_j,q_i)}
$$
$$
= \frac{-Q(p_j)\prod_{n=1}^{6}\theta(a_n)}{a(p_j)d(-p_j)a(-p_j)d(p_j)} \sum_{\sigma=\pm} \frac{\sigma\mathbf{A}_\varepsilon(\sigma p_j)Q(p_j - \sigma\eta)}{\theta(2p_j + \sigma\eta)\,\theta_s(p_j - \sigma\eta, q_i)\,\theta_s(p_j,q_i)} . \tag{117}
$$

Using the identity

$$
\frac{\theta'(u-v)}{\theta(u-v)} - \frac{\theta'(u-v\pm\eta)}{\theta(u-v\pm\eta)} - \frac{\theta'(u+v)}{\theta(u+v)} + \frac{\theta'(u+v\pm\eta)}{\theta(u+v\pm\eta)} = \frac{\theta(2u\pm\eta)\,\theta(2v)\,\theta(\pm\eta)\,\theta'(0)}{\theta_s(u,v)\,\theta_s(u\pm\eta,v)} , \tag{118}
$$

which can be proven by usual arguments concerning elliptic functions, we obtain that

$$
\left[\mathcal{G}^{(1,1)}\right]_{i,j} = \frac{Q(p_j)\prod_{n=1}^{6}\theta(a_n)}{a(p_j)d(-p_j)a(-p_j)d(p_j)} \frac{\left[\mathcal{S}_Q(\mathbf{p},\mathbf{q})\right]_{j,i}}{\theta'(0)\,\theta(\eta)\,\theta(2q_i)\,\theta_s(2p_j,\eta)} , \tag{119}
$$

in which $\left[\mathcal{S}_Q(\mathbf{p},\mathbf{q})\right]_{i,j}$ is given by the expression (84). Noticing that

$$
\prod_{j=1}^{M}\left[a(p_j)d(-p_j)a(-p_j)d(p_j)\right] = \prod_{n=1}^{N}\left[P(\xi_n - \eta/2)P(\xi_n + \eta/2)\right], \tag{120}
$$

we finally obtain

$$
\det_M\left[\mathcal{G}^{(1,1)}\right] = \prod_{j=1}^{M} \frac{Q(p_j)\prod_{n=1}^{6}\theta(a_n)}{\theta'(0)\,\theta(\eta)\,\theta(2q_i)\,\theta_s(2p_j,\eta)} \frac{\det_M\left[\mathcal{S}_Q(\mathbf{p},\mathbf{q})\right]}{\prod_{n=1}^{N}P(\xi_n - \frac{\eta}{2})P(\xi_n + \frac{\eta}{2})} . \tag{121}
$$

Therefore, we have proven that

$$
{}_\varepsilon\langle P|Q\rangle_\varepsilon = \frac{\mathsf{c}_Q\,\det_M\left[\mathcal{S}_Q(\mathbf{p},\mathbf{q})\right]}{V(p_1,\ldots,p_M)\prod_{j=1}^{M}\theta_s(2p_j,\eta)} , \tag{122}
$$

in which

$$
\mathsf{c}_Q = \prod_{n=1}^{N} \frac{Q(\xi_n - \frac{\eta}{2})Q(\xi_n + \frac{\eta}{2})}{\prod_{i=1}^{6}\theta(\xi_n - a_i)} \prod_{j=1}^{M} \frac{\prod_{i=1}^{6}\theta(a_i)}{\theta'(0)\,\theta(\eta)\,\theta(2q_j)} \frac{\det_M\left[\mathcal{G}^{(2,2)}\right]}{V(q_1,\ldots,q_M)\,V(\xi_1,\ldots,\xi_N)\,\det_N\mathcal{X}} \tag{123}
$$

is a coefficient which does not depend on $P$ (note also that it should be independent of $\gamma$). It means that, if $\widetilde{P}$ is another even elliptic polynomial of the form (82) and $\tilde{\mathbf{p}} = (\tilde{p}_1,\ldots,\tilde{p}_M)$ is its $M$-tuple of roots, and if ${}_\varepsilon\langle\widetilde{P}|$ is the corresponding left separate state built from $\widetilde{P}$ as in (71), we have

$$
{}_\varepsilon\langle\widetilde{P}|Q\rangle_\varepsilon = \frac{\mathsf{c}_Q\,\det_M\left[\mathcal{S}_Q(\tilde{\mathbf{p}},\mathbf{q})\right]}{V(\tilde{p}_1,\ldots,\tilde{p}_M)\prod_{j=1}^{M}\theta_s(2\tilde{p}_j,\eta)} , \tag{124}
$$

with the same coefficient $\mathsf{c}_Q$ (123), so that

$$
\frac{{}_\varepsilon\langle P|Q\rangle_\varepsilon}{{}_\varepsilon\langle\widetilde{P}|Q\rangle_\varepsilon} = \frac{V(\tilde{p}_1,\ldots,\tilde{p}_M)}{V(p_1,\ldots,p_M)} \prod_{j=1}^{M} \frac{\theta_s(2\tilde{p}_j,\eta)}{\theta_s(2p_j,\eta)} \frac{\det_M\left[\mathcal{S}_Q(\mathbf{p},\mathbf{q})\right]}{\det_M\left[\mathcal{S}_Q(\tilde{\mathbf{p}},\mathbf{q})\right]} . \tag{125}
$$

In the limit $\tilde{p}_j \to q_j$, $1 \le j \le M$, we obtain (83).

To conclude this section, let us briefly comment about the case in which 2 of the 6 boundary parameters are opposite from each others, say for instance $a_5 = -a_6$. We have already mentioned that, in this special case, it was possible to apply the strategy of [4,5] to compute the scalar products, see the footnote 8. The choice $a_5 = -a_6$ also induces some simplifications in the new approach presented in this section: under this hypothesis, we can indeed simply choose $\gamma = a_5$ in (101)-(105), so that the functions (105) no longer have a pole in $\gamma$. This means that the terms of the form $\mathsf{X}_\gamma(u)\,\mathsf{M}_{\{a\}}(\gamma,v)$ in the expressions (112)-(113) vanish. Therefore, in particular, the matrix $\mathcal{G}^{(2,2)}$ (113) is simply a diagonal matrix, instead of being the sum of a diagonal matrix and a rank one matrix, so that the explicit expression of its determinant – and hence of the constant $c_Q$ (123) – is much simpler. The general formula issued from (115), in which we do not suppose $q_1,\ldots,q_M$ to be a solution of the Bethe equations, would also be much less cumbersome in that case. However, this does not induce any particular simplification in our main result (83), once we impose the Bethe equations for $q_1,\ldots,q_M$ and normalise adequately the scalar product.

## 5.2 Proof of theorem 4.1 in the case $2M > N$

Let us now explain how one can adapt the computation presented in the last section to the case $2M > N$. As in [3–5], we need to change the size of the scalar product determinant representation since the number of roots of $P$ and $Q$ (82) is not equal to the number $N$ of inhomogeneity parameters. In the present case, this can be done by means of the following identities.

**Lemma 5.1.** *Let $x_1,\ldots,x_L$ be generic parameters. Let $\epsilon = \pm 1$, and let $r_1,\ldots,r_\ell$ be such that*

$$\epsilon r_j \ne \pm r_i, \pm r_i - \eta \quad \mathrm{mod}\,(\pi,\pi\omega), \qquad \forall i < j\,. \tag{126}$$

*Then, for any function $F$ which is regular in $\frac{\eta}{2} + \epsilon r_j$, $1 \le j \le \ell$, any basis $\{\Theta_L^{(j)}\}_{1 \le j \le L}$ of $\Xi_L$, any basis $\{\Theta_{L+\ell}^{(j)}\}_{1 \le j \le L+\ell}$ of $\Xi_{L+\ell}$, we have*

$$
\frac{\det_{1 \le i,j \le L}\left[\Theta_L^{(j)}(x_i - \frac{\eta}{2}) - F(x_i)\Theta_L^{(j)}(x_i + \frac{\eta}{2})\right]}{\det_{1 \le i,j \le L}\left[\Theta_L^{(j)}(x_i - \frac{\eta}{2})\right]}
$$
$$
= \lim_{x_{L+1} \to \frac{\eta}{2}+\epsilon r_1} \cdots \lim_{x_{L+\ell} \to \frac{\eta}{2}+\epsilon r_\ell} \frac{\det_{1 \le i,j \le L+\ell}\left[\Theta_{L+\ell}^{(j)}(x_i - \frac{\eta}{2}) - F_{\{r\}}^{(\ell)}(x_i)\Theta_{L+\ell}^{(j)}(x_i + \frac{\eta}{2})\right]}{\det_{1 \le i,j \le L+\ell}\left[\Theta_{L+\ell}^{(j)}(x_i - \frac{\eta}{2})\right]}\,,
\tag{127}
$$

*in which*

$$F_{\{r\}}^{(\ell)}(\lambda) = F(\lambda)\prod_{j=1}^{\ell} \frac{\theta_s(\lambda - \frac{\eta}{2}, r_j)}{\theta_s(\lambda + \frac{\eta}{2}, r_j)}\,. \tag{128}$$

*Proof.* We proceed by induction on $\ell$. (127) obviously holds for $\ell = 0$. Let us suppose that it holds for a given $\ell \ge 0$, and let us consider a basis of $\Xi_{L+\ell+1}$ of the form

$$\Theta_{L+\ell+1}^{(j)}(\lambda) = \frac{Y_{L+\ell+1}(\lambda)}{\theta_s(\lambda, y_j)}\,, \quad 1 \le j \le L+\ell+1\,, \quad \text{with} \quad Y_{L+\ell+1}(\lambda) = \prod_{k=1}^{L+\ell+1} \theta_s(\lambda, y_k)\,, \tag{129}$$

in which we suppose that the parameters $\pm y_1,\ldots,\pm y_{L+\ell+1}$ are pairwise distinct modulo $\pi,\pi\omega$, with the particular choice $y_{L+\ell+1} = r_{\ell+1}$. Then the functions

$$\Theta_{L+\ell}^{(j)}(\lambda) = \frac{\Theta_{L+\ell+1}^{(j)}(\lambda)}{\theta_s(\lambda, r_{\ell+1})}\,, \quad 1 \le j \le L+\ell\,, \tag{130}$$

define a basis of $\Xi_{L+\ell}$, so that, with this choice of basis in the right hand side of (127),

$$
\begin{aligned}
&\frac{\det_{1\leq i,j\leq L+\ell}\left[\Theta_{L+\ell}^{(j)}(x_i-\tfrac{\eta}{2})-F_{\{r\}}^{(\ell)}(x_i)\Theta_{L+\ell}^{(j)}(x_i+\tfrac{\eta}{2})\right]}{\det_{1\leq i,j\leq L+\ell}\left[\Theta_{L+\ell}^{(j)}(x_i-\tfrac{\eta}{2})\right]}\\
&=\frac{\det_{1\leq i,j\leq L+\ell}\left[\frac{\Theta_{L+\ell+1}^{(j)}(x_i-\tfrac{\eta}{2})}{\theta_s(x_i-\tfrac{\eta}{2},r_{\ell+1})}-F_{\{r\}}^{(\ell)}(x_i)\frac{\Theta_{L+\ell+1}^{(j)}(x_i+\tfrac{\eta}{2})}{\theta_s(x_i+\tfrac{\eta}{2},r_{\ell+1})}\right]}{\det_{1\leq i,j\leq L+\ell}\left[\frac{\Theta_{L+\ell+1}^{(j)}(x_i-\tfrac{\eta}{2})}{\theta_s(x_i-\tfrac{\eta}{2},r_{\ell+1})}\right]}\\
&=\frac{\det_{1\leq i,j\leq L+\ell}\left[\Theta_{L+\ell+1}^{(j)}(x_i-\tfrac{\eta}{2})-F_{\{r\}}^{(\ell)}(x_i)\frac{\theta_s(x_i-\tfrac{\eta}{2},r_{\ell+1})}{\theta_s(x_i+\tfrac{\eta}{2},r_{\ell+1})}\Theta_{L+\ell+1}^{(j)}(x_i+\tfrac{\eta}{2})\right]}{\det_{1\leq i,j\leq L+\ell}\left[\Theta_{L+\ell+1}^{(j)}(x_i-\tfrac{\eta}{2})\right]}\\
&=\lim_{x_{L+\ell+1}\to\epsilon r_{\ell+1}+\tfrac{\eta}{2}}\frac{\det_{1\leq i,j\leq L+\ell+1}\left[\Theta_{L+\ell+1}^{(j)}(x_i-\tfrac{\eta}{2})-F_{\{r\}}^{(\ell+1)}(x_i)\Theta_{L+\ell+1}^{(j)}(x_i+\tfrac{\eta}{2})\right]}{\det_{1\leq i,j\leq L+\ell+1}\left[\Theta_{L+\ell+1}^{(j)}(x_i-\tfrac{\eta}{2})\right]}.
\end{aligned}
$$
(131)

In the last equality we have used that $\Theta_{L+\ell+1}^{(j)}(\epsilon r_{\ell+1}) = 0$ if $j \neq L+\ell+1$, and that $F_{\{r\}}^{(\ell+1)}(\epsilon r_{\ell+1}+\tfrac{\eta}{2}) = 0$ ($F_{\{r\}}^{(\ell)}$ being regular at $\tfrac{\eta}{2}+\epsilon r_{\ell+1}$ due to (126)), so that only the last coefficient of the last line of each determinant survives in the limit. This proves the statement for $\ell+1$. $\qquad\square$

We now formulate a special case of Lemma 5.1 that will be useful for the computation of the scalar products.

**Lemma 5.2.** *Let $x_1,\ldots,x_L$ be generic parameters, let $a \in \mathbb{C}$, and let $F$ be a function which is regular at the points $-a+j\eta$, $1 \leq j \leq \ell$, for a given $\ell \in \mathbb{N}$. Then, for any basis $\{\Theta_L^{(j)}\}_{1\leq j\leq L}$ of $\Xi_L$, any basis $\{\Theta_{L+\ell}^{(j)}\}_{1\leq j\leq L+\ell}$ of $\Xi_{L+\ell}$, we have*

$$
\begin{aligned}
&\frac{\det_{1\leq i,j\leq L}\left[\Theta_L^{(j)}(x_i-\tfrac{\eta}{2})-F(x_i)\frac{\theta(x_i+a)}{\theta(x_i-a)}\Theta_L^{(j)}(x_i+\tfrac{\eta}{2})\right]}{\det_{1\leq i,j\leq L}\left[\Theta_L^{(j)}(x_i-\tfrac{\eta}{2})\right]}\\
&=\lim_{x_{L+1}\to\eta-a}\cdots\lim_{x_{L+\ell}\to\ell\eta-a}\frac{\det_{1\leq i,j\leq L+\ell}\left[\Theta_{L+\ell}^{(j)}(x_i-\tfrac{\eta}{2})-F(x_i)\frac{\theta(x_i+a-\ell\eta)}{\theta(x_i-a+\ell\eta)}\Theta_{L+\ell}^{(j)}(x_i+\tfrac{\eta}{2})\right]}{\det_{1\leq i,j\leq L+\ell}\left[\Theta_{L+\ell}^{(j)}(x_i-\tfrac{\eta}{2})\right]}.
\end{aligned}
$$
(132)

*Proof.* It is enough to set $r_j = j\eta - \tfrac{\eta}{2} - a$ in the hypothesis of Lemma 5.1. These parameters satisfy (126), and

$$
\begin{aligned}
\frac{\theta(\lambda+a)}{\theta(\lambda-a)}\prod_{j=1}^{\ell}\frac{\theta_s(\lambda-\tfrac{\eta}{2},r_j)}{\theta_s(\lambda+\tfrac{\eta}{2},r_j)} &= \frac{\theta(\lambda+a)}{\theta(\lambda-a)}\prod_{j=1}^{\ell}\frac{\theta(\lambda-a+(j-1)\eta)\,\theta(\lambda+a-j\eta)}{\theta(\lambda-a+j\eta)\,\theta(\lambda+a-(j-1)\eta)}\\
&= \frac{\theta(\lambda+a-\ell\eta)}{\theta(\lambda-a+\ell\eta)},
\end{aligned}
$$
(133)

which leads to (132). $\qquad\square$

These results can directly be applied to transform the original SoV representation (78) of the scalar product of the separate states $_\varepsilon\langle P|$ and $|Q\rangle_\varepsilon$, with $P$ and $Q$ of the form (82) if

$2M = N + \ell$, $\ell \geq 0$. With the notations (90), it follows from Lemma 5.2 that

$$
\begin{aligned}
{}_{\varepsilon}\langle P \mid Q \rangle_{\varepsilon} &= \prod_{n=1}^{N} (PQ)(\xi_n^{(0)}) \frac{\det_{1 \leq i,j \leq N}\left[\Theta_N^{(j)}(\xi_i^{(1)})\right]}{\det_{1 \leq i,j \leq N}\left[\Theta_N^{(j)}(\xi_i)\right]} \\
&\quad \times \frac{\det_{1 \leq i,j \leq N}\left[\Theta_N^{(j)}(\xi_i^{(1)}) - \prod_{n=1}^{6} \frac{\theta(\xi_i + a_n)}{\theta(\xi_i - a_n)} \frac{(PQ)(\xi_i^{(1)})}{(PQ)(\xi_i^{(0)})} \Theta_N^{(j)}(\xi_i^{(0)})\right]}{\det_{1 \leq i,j \leq N}\left[\Theta_N^{(j)}(\xi_i^{(1)})\right]} \\
&= \prod_{n=1}^{N} (PQ)(\xi_n^{(0)}) \frac{V(\xi_1^{(1)}, \ldots, \xi_N^{(1)})}{V(\xi_1, \ldots, \xi_N)} \\
&\quad \times \lim_{\xi_{N+1} \to \eta - a_s} \cdots \lim_{\xi_{N+\ell} \to \ell\eta - a_s} \frac{\det_{1 \leq i,j \leq 2M}\left[\Theta_{2M}^{(j)}(\xi_i^{(1)}) - \prod_{n=1}^{6} \frac{\theta(\xi_i + a_n')}{\theta(\xi_i - a_n')} \frac{(PQ)(\xi_i^{(1)})}{(PQ)(\xi_i^{(0)})} \Theta_{2M}^{(j)}(\xi_i^{(0)})\right]}{\det_{1 \leq i,j \leq 2M}\left[\Theta_{2M}^{(j)}(\xi_i^{(1)})\right]},
\end{aligned}
\tag{134}
$$

for some given $a_s \in \{a_n\}_{1 \leq n \leq 6}$, and in which we have set

$$
a_n' = \begin{cases} a_s - (2M - N)\eta, & \text{if } n = s, \\ a_n, & \text{otherwise.} \end{cases}
\tag{135}
$$

Note that these parameters satisfy the sum rule

$$
\sum_{\ell=1}^{6} a_\ell' = \eta.
\tag{136}
$$

We can now transform the determinant in the numerator of the last line of (134) according to the process presented in the last section. However, so as to avoid dealing with complicated coefficients which are irrelevant for the computation of the normalised scalar product (83), let us directly consider the ratio of the scalar product ${}_{\varepsilon}\langle P \mid Q \rangle_{\varepsilon}$ by the scalar product ${}_{\varepsilon}\langle \widetilde{P} \mid Q \rangle_{\varepsilon}$, in which $\widetilde{P}$ is another function of the form (82) with roots $\tilde{p}_1, \ldots, \tilde{p}_M$:

$$
\frac{{}_{\varepsilon}\langle P \mid Q \rangle_{\varepsilon}}{{}_{\varepsilon}\langle \widetilde{P} \mid Q \rangle_{\varepsilon}} = \prod_{n=1}^{N} \frac{P(\xi_n^{(0)})}{\widetilde{P}(\xi_n^{(0)})} \frac{\det_{1 \leq i,j \leq 2M}\left[\frac{1}{\theta_s(\xi_i - \frac{\eta}{2}, \tilde{w}_j)}\right]}{\det_{1 \leq i,j \leq 2M}\left[\frac{1}{\theta_s(\xi_i - \frac{\eta}{2}, w_j)}\right]} \frac{\det_{1 \leq i,j \leq 2M}\left[\mathsf{M}_{\{a'\}}(\xi_i, w_j)\right]}{\det_{1 \leq i,j \leq 2M}\left[\mathsf{M}_{\{a'\}}(\xi_i, \tilde{w}_j)\right]},
\tag{137}
$$

in which we have denoted, similarly as in (94),

$$
(w_1, \ldots, w_{2M}) = (p_1, \ldots, p_M, q_1, \ldots, q_M),
\tag{138}
$$
$$
(\tilde{w}_1, \ldots, \tilde{w}_{2M}) = (\tilde{p}_1, \ldots, \tilde{p}_M, q_1, \ldots, q_M),
\tag{139}
$$

and in which we have set

$$
\xi_{N+n} = n\eta - a_s, \qquad 1 \leq n \leq 2M - N.
\tag{140}
$$

The function $\mathsf{M}_{\{a'\}}(u, v)$ is defined as in (98), in which the parameters $a_1, \ldots, a_6$ are replaced by $a_1', \ldots, a_6'$.

The ratio of generalised Cauchy determinants in (137) can easily be computed:

$$
\begin{aligned}
\frac{\det_{1 \leq i,j \leq 2M}\left[\frac{1}{\theta_s(\xi_i - \frac{\eta}{2}, \tilde{w}_j)}\right]}{\det_{1 \leq i,j \leq 2M}\left[\frac{1}{\theta_s(\xi_i - \frac{\eta}{2}, w_j)}\right]} &= \frac{V(\tilde{w}_1, \ldots, \tilde{w}_{2M})}{V(w_1, \ldots, w_{2M})} \prod_{i,j=1}^{2M} \frac{\theta_s(\xi_i - \frac{\eta}{2}, w_j)}{\theta_s(\xi_i - \frac{\eta}{2}, \tilde{w}_j)} \\
&= \frac{V(\tilde{p}_1, \ldots, \tilde{p}_M)}{V(p_1, \ldots, p_M)} \prod_{j=1}^{M} \frac{Q(\tilde{p}_j)}{Q(p_j)} \prod_{n=1}^{N} \frac{P(\xi_n - \frac{\eta}{2})}{\widetilde{P}(\xi_n - \frac{\eta}{2})} \prod_{n=1}^{2M-N} \frac{P(n\eta - \frac{\eta}{2} - a_s)}{\widetilde{P}(n\eta - \frac{\eta}{2} - a_s)}.
\end{aligned}
\tag{141}
$$

To transform the last ratio of determinants in (137), we can now define, in terms of the extended set of parameters $\xi_1, \ldots, \xi_{2M}$, a basis $\{\bar{\Theta}_{2M}^{(j)}\}_{1 \leq j \leq 2M}$ of $\Xi_{2M}$ as in (101) and a $2M \times 2M$ matrix $\mathcal{X}$ as in (103). Multiplying both matrices in the numerator and denominator of the last line of (137), we get

$$\frac{\det_{1 \leq i,j \leq 2M}\left[\mathsf{M}_{\{a'\}}(\xi_i, w_j)\right]}{\det_{1 \leq i,j \leq 2M}\left[\mathsf{M}_{\{a'\}}(\xi_i, \tilde{w}_j)\right]} = \frac{\det_{2M} \mathcal{G}}{\det_{2M} \widetilde{\mathcal{G}}}, \tag{142}$$

in which $\mathcal{G}$ and $\widetilde{\mathcal{G}}$ have the following block form:

$$\mathcal{G} = \begin{pmatrix} \mathcal{G}^{(1,1)} & \mathcal{G}^{(1,2)} \\ \mathcal{G}^{(2,1)} & \mathcal{G}^{(2,2)} \end{pmatrix}, \qquad \widetilde{\mathcal{G}} = \begin{pmatrix} \widetilde{\mathcal{G}}^{(1,1)} & \widetilde{\mathcal{G}}^{(1,2)} \\ \widetilde{\mathcal{G}}^{(2,1)} & \widetilde{\mathcal{G}}^{(2,2)} \end{pmatrix}. \tag{143}$$

Here the blocks $\mathcal{G}^{(i,j)}$ and $\widetilde{\mathcal{G}}^{(i,j)}$, $1 \leq i, j \leq 2$, are of size $M \times M$, their elements can be computed as in the last section, in terms of the residues of the functions

$$\mathsf{F}_{i,j}(\lambda) \equiv \mathsf{F}_{Q,\{a'\},\{\xi\}}(\lambda; z_i, \tilde{z}_i, w_j), \tag{144}$$

defined as in (105), but in terms of the parameters $\{a_i'\}_{1 \leq i \leq 6}$ (135) instead of $\{a_i\}_{1 \leq i \leq 6}$, and in terms of the extended set of inhomogeneity parameters $\xi_1, \ldots, \xi_{2M}$. Note that, thanks to (136), the functions (144) still satisfy the properties (107) and (108). Hence, the elements of the blocks $\mathcal{G}^{(i,j)}$, $1 \leq i, j \leq 2$, are given by the expressions (110)-(113) in terms of the parameters $\{a_i'\}_{1 \leq i \leq 6}$ (135) instead of $\{a_i\}_{1 \leq i \leq 6}$, and with the functions $a$ and $d$ (23) replaced respectively by

$$a_\xi(\lambda) = \prod_{n=1}^{2M} \theta(\lambda - \xi_n + \eta/2) = a(\lambda) \prod_{n=1}^{2M-N} \theta(\lambda - n\eta + a_s + \eta/2), \tag{145}$$

$$d_\xi(\lambda) = a_\xi(\lambda - \eta) = d(\lambda) \prod_{n=1}^{2M-N} \theta(\lambda - n\eta + a_s - \eta/2). \tag{146}$$

The elements of the blocks $\widetilde{\mathcal{G}}^{(i,j)}$, $1 \leq i, j \leq 2$, are given by similar expressions, however in terms of the roots $\tilde{p}_1, \ldots, \tilde{p}_M$ instead of $p_1, \ldots, p_M$. Note that

$$\widetilde{\mathcal{G}}^{(i,2)} = \mathcal{G}^{(i,2)}, \qquad 1 \leq i \leq 2, \tag{147}$$

since the corresponding columns do not depend on the roots of $P$ (or $\widetilde{P}$).

In particular, the elements of the block $\mathcal{G}^{(1,2)} = \widetilde{\mathcal{G}}^{(1,2)}$ are, from (111),

$$
\begin{aligned}
\left[\mathcal{G}^{(1,2)}\right]_{i,j} = \left[\widetilde{\mathcal{G}}^{(1,2)}\right]_{i,j} &= \delta_{i,j} \frac{Q_i(q_i)}{\theta(2q_i)} \sum_{\sigma=\pm} \frac{\sigma Q_i(q_i + \sigma\eta) \prod_{n=1}^6 \theta(\sigma q_i + \frac{\eta}{2} - a_n')}{a_\xi(\sigma q_i) d_\xi(-\sigma q_i)} \\
&= \delta_{i,j} \frac{Q_i(q_i - \eta) Q_i(q_i) \prod_{n=1}^6 \theta(q_i - \frac{\eta}{2} + a_n')}{a_\xi(-q_i) d_\xi(q_i)} \frac{}{\theta(2q_i)} \\
&\quad \times \left[ \frac{a(-q_i) d(q_i)}{a(q_i) d(-q_i)} \frac{Q_i(q_i + \eta)}{Q_i(q_i - \eta)} \prod_{n=1}^6 \frac{\theta(q_i + \frac{\eta}{2} - a_n)}{\theta(q_i - \frac{\eta}{2} + a_n)} - 1 \right],
\end{aligned}
\tag{148}
$$

in which we have used that

$$\prod_{n=1}^6 \frac{\theta(\lambda + \frac{\eta}{2} - a_n')}{\theta(\lambda - \frac{\eta}{2} + a_n')} \prod_{n=1}^{2M-N} \frac{\theta_s(\lambda - \frac{\eta}{2}, n\eta - a_s)}{\theta_s(\lambda + \frac{\eta}{2}, n\eta - a_s)} = \prod_{n=1}^6 \frac{\theta(\lambda + \frac{\eta}{2} - a_n)}{\theta(\lambda - \frac{\eta}{2} + a_n)}. \tag{149}$$

Hence, if we now suppose that the roots $q_1, \ldots, q_M$ of $Q$ satisfy the Bethe equations (92), the blocks $\mathcal{G}^{(1,2)}$ and $\widetilde{\mathcal{G}}^{(1,2)}$ vanish, so that

$$\frac{\det_{1 \le i,j \le 2M} \left[ M_{\{a'\}}(\xi_i, w_j) \right]}{\det_{1 \le i,j \le 2M} \left[ M_{\{a'\}}(\xi_i, \tilde{w}_j) \right]} = \frac{\det_M \mathcal{G}^{(1,1)}}{\det_M \widetilde{\mathcal{G}}^{(1,1)}} . \tag{150}$$

The matrix elements of $\mathcal{G}^{(1,1)}$ are, from (110),

$$
\begin{aligned}
\left[ \mathcal{G}^{(1,1)} \right]_{i,j} &= \frac{Q_i(p_j)}{\theta(2p_j)} \sum_{\sigma=\pm} \frac{\sigma Q_i(p_j + \sigma\eta) \prod_{n=1}^6 \theta(\sigma p_j + \frac{\eta}{2} - a'_n)}{a_\xi(\sigma p_j) d_\xi(-\sigma p_j)} \\
&= \frac{Q_i(p_j - \eta) Q_i(p_j)}{(-1)^N a(-p_j) d(p_j)} \frac{\prod_{n=1}^6 \theta(p_j - \frac{\eta}{2} + a'_n)}{\theta(2p_j) \prod_{n=1}^{2M-N} \theta_s(p_j - \frac{\eta}{2}, n\eta - a_s)} \\
&\quad \times \left[ \frac{a(-p_j) d(p_j)}{a(p_j) d(-p_j)} \frac{Q_i(p_j + \eta)}{Q_i(p_j - \eta)} \prod_{n=1}^6 \frac{\theta(p_j + \frac{\eta}{2} - a_n)}{\theta(p_j - \frac{\eta}{2} + a_n)} - 1 \right] \\
&= \frac{-Q(p_j) \prod_{n=1}^6 \theta(a_n)}{a(p_j) d(-p_j) a(-p_j) d(p_j)} \prod_{n=1}^{2M-N} \frac{1}{\theta_s(p_j, n\eta - a_s - \frac{\eta}{2})} \\
&\quad \times \sum_{\sigma=\pm} \frac{\sigma A_\varepsilon(\sigma p_j) Q(p_j - \sigma\eta)}{\theta(2p_j + \sigma\eta) \theta_s(p_j - \sigma\eta, q_i) \theta_s(p_j, q_i)} ,
\end{aligned}
\tag{151}
$$

in which we have once again used (149). The elements of $\widetilde{\mathcal{G}}^{(1,1)}$ have similar expressions in terms of $\tilde{p}_j$ instead of $p_j$.

The remaining part of the computation is then completely similar as in the previous section, so that

$$
\begin{aligned}
\frac{\det_{1 \le i,j \le 2M} \left[ M_{\{a'\}}(\xi_i, w_j) \right]}{\det_{1 \le i,j \le 2M} \left[ M_{\{a'\}}(\xi_i, \tilde{w}_j) \right]} &= \prod_{n=1}^N \frac{\widetilde{P}(\xi_n + \frac{\eta}{2}) \widetilde{P}(\xi_n - \frac{\eta}{2})}{P(\xi_n + \frac{\eta}{2}) P(\xi_n - \frac{\eta}{2})} \prod_{n=1}^{2M-N} \frac{\widetilde{P}(n\eta - a_s - \frac{\eta}{2})}{P(n\eta - a_s - \frac{\eta}{2})} \\
&\quad \times \prod_{j=1}^M \frac{\theta_s(2\tilde{p}_j, \eta) Q(p_j)}{\theta_s(2p_j, \eta) Q(\tilde{p}_j)} \frac{\det_M \mathcal{S}_Q(p_j, q_i)}{\det_M \mathcal{S}_Q(\tilde{p}_j, q_i)} ,
\end{aligned}
\tag{152}
$$

and, from (137), (141) and (152), we get the result.

## 5.3 Proof of theorem 4.1 in the case $2M < N$

Finally, to conclude the proof of Theorem 4.1, let us consider the case $2M < N$.

For generic $p_1, \ldots, p_M, q_1, \ldots, q_M, \xi_1, \ldots, \xi_N$, we consider (78), in which we choose a basis of the form (81) with $(w_1, \ldots, w_{2M}) = (p_1, \ldots, p_M, q_1, \ldots, q_M)$, the other variables $w_{2M+1}, \ldots, w_N$ being still to be specified, i.e.

$$W(\lambda) = P(\lambda) Q(\lambda) W_R(\lambda), \qquad \text{with} \quad W_R(\lambda) = \prod_{n=1}^{N-2M} \theta_s(\lambda, w_{2M+n}). \tag{153}$$

Then

$$
{}_\varepsilon\langle P | Q \rangle_\varepsilon = \prod_{n=1}^N \frac{(PQ)(\xi_n^{(0)}) W(\xi_n^{(1)})}{W(\xi_n)} \frac{\det_{1 \le i,j \le N} \left[ \frac{1}{\theta_s(\xi_i^{(1)}, w_j)} - \prod_{n=1}^6 \frac{\theta(a_n + \xi_i)}{\theta(a_n - \xi_i)} \frac{W_R(\xi_i^{(0)})}{W_R(\xi_i^{(1)})} \frac{1}{\theta_s(\xi_i^{(0)}, w_j)} \right]}{\det_{1 \le i,j \le N} \left[ \frac{1}{\theta_s(\xi_i, w_j)} \right]} . \tag{154}
$$

It is possible to reduce the above determinant to a determinant of the form (97)-(98) by conveniently setting, for some $s \in \{1, \ldots, 6\}$, the additional variables $w_{2M+j}$ as

$$w_{2M+j} = a_s + j\eta - \frac{\eta}{2}, \qquad 1 \le j \le N - 2M, \tag{155}$$

so that

$$\frac{\theta(\lambda + a_s)}{\theta(\lambda - a_s)} \frac{W_R(\lambda + \frac{\eta}{2})}{W_R(\lambda - \frac{\eta}{2})} = \frac{\theta(\lambda + a_s + (N - 2M)\eta)}{\theta(\lambda - a_s - (N - 2M)\eta)}. \tag{156}$$

Defining

$$a'_n = \begin{cases} a_s + (N - 2M)\eta, & \text{if } n = s, \\ a_n, & \text{otherwise,} \end{cases} \tag{157}$$

we therefore obtain

$$\begin{aligned}
_\varepsilon \langle P | Q \rangle_\varepsilon &= \prod_{n=1}^{N} \frac{-(PQ)(\xi_n^{(0)}) W(\xi_n^{(1)})}{W(\xi_n) \prod_{l=1}^{6} \theta(a'_l - \xi_n)} \frac{\det_{1 \le i,j \le N}\left[ \mathsf{M}_{\{a'\}}(\xi_i, w_j) \right]}{\det_{1 \le i,j \le N}\left[ \frac{1}{\theta_s(\xi_i, w_j)} \right]} \\
&= \prod_{n=1}^{N} \frac{(PQ)(\xi_n^{(0)}) W(\xi_n^{(1)})}{-\prod_{l=1}^{6} \theta(a'_l - \xi_n)} \frac{\det_{1 \le i,j \le N}\left[ \mathsf{M}_{\{a'\}}(\xi_i, w_j) \right]}{V(w_1, \ldots, w_N) V(\xi_N, \ldots, \xi_1)}.
\end{aligned} \tag{158}$$

Note that the parameters $a'_n$, $1 \le n \le 6$, defined by (157) also satisfy the sum rule (136).

We can now transform the determinant of the matrix $\mathcal{M} \equiv (\mathsf{M}_{\{a'\}}(\xi_i, w_j))_{1 \le i,j \le N}$ according to the previous approach, but we have to adapt slightly this approach since now $2M < N$. We define the following generalisations of the functions (105):

$$\begin{aligned}
\mathsf{F}_{i,j}(\lambda) &\equiv \mathsf{F}_{Q,\{a'\},\{\xi\},\{r\}}(\lambda; z_i, \tilde{z}_i, w_j) \\
&= \frac{\theta'(0) Q(\lambda + \frac{\eta}{2}) Q(\lambda - \frac{\eta}{2}) R(\lambda)}{\theta_s(\lambda, z_i) \theta_s(\lambda, \tilde{z}_i) \prod_{n=1}^{N} \theta_s(\lambda, \xi_n)} \sum_{\epsilon = \pm} \epsilon \frac{\prod_{n=1}^{6} \theta(\lambda + \epsilon a'_n)}{\theta_s(\lambda + \epsilon \frac{\eta}{2}, w_j)} \\
&= \frac{\theta'(0) Q(\lambda + \frac{\eta}{2}) Q(\lambda - \frac{\eta}{2})}{\theta_s(\lambda, z_i) \theta_s(\lambda, \tilde{z}_i) \prod_{n=1}^{N} \theta_s(\lambda, \xi_n)} \sum_{\epsilon = \pm} \epsilon \frac{W_R(\lambda + \epsilon \frac{\eta}{2}) \prod_{n=1}^{6} \theta(\lambda + \epsilon a_n)}{\theta_s(\lambda + \epsilon \frac{\eta}{2}, w_j)},
\end{aligned} \tag{159}$$

in which we have set

$$R(\lambda) = \prod_{n=1}^{N-2M} \theta_s(\lambda, r_n), \quad \text{with} \quad r_n = w_{2M+n} - \frac{\eta}{2} = a_s + (n-1)\eta, \quad 1 \le n \le N - 2M, \tag{160}$$

and

$$(z_i, \tilde{z}_i) = \begin{cases} (q_i + \frac{\eta}{2}, q_i - \frac{\eta}{2}), & \text{if } i \le M, \\ (q_{i-M} + \frac{\eta}{2}, \gamma), & \text{if } M < i \le 2M, \\ (r_{i-2M}, \gamma), & \text{if } 2M < i \le N, \end{cases} \tag{161}$$

with $\gamma$ being an arbitrary generic number. In (159), we have used that, with the choice (160) for the additional variables $r_1, \ldots, r_{N-2M}$,

$$R(\lambda) \theta(\lambda \pm a'_s) = W_R(\lambda \pm \eta/2) \theta(\lambda \pm a_s). \tag{162}$$

The functions (159) satisfy the properties (107) and (108).

Let now $\mathcal{X}$ be the matrix with elements

$$\mathcal{X}_{i,k} = \frac{\bar{\Theta}_N^{(i)}(\xi_k)}{\theta_s(\xi_k,\gamma)\,\theta(2\xi_k)\prod_{n\neq k}\theta_s(\xi_k,\xi_n)}, \qquad 1 \le i,k \le N, \tag{163}$$

in which $\{\bar{\Theta}_N^{(j)}\}_{1\le j\le N}$ is the basis of $\Xi_N$ defined by

$$\bar{\Theta}_N^{(j)}(\lambda) = \frac{Q(\lambda+\frac{\eta}{2})Q(\lambda-\frac{\eta}{2})R(\lambda)\,\theta_s(\lambda,\gamma)}{\theta_s(\lambda,z_j)\,\theta_s(\lambda,\tilde{z}_j)}, \qquad 1 \le j \le N, \tag{164}$$

and let us compute the product of the matrix $\mathcal{X}$ by the matrix $\mathcal{M}$. It takes the following block form, which generalises (109):

$$\mathcal{X}\cdot\mathcal{M} = \left(\sum_{k=1}^N \mathrm{Res}(\mathsf{F}_{i,j};\xi_k)\right)_{1\le i,j\le N} = \begin{pmatrix} \mathcal{G}^{(1,1)} & \mathcal{G}^{(1,2)} & \mathcal{G}^{(1,3)} \\ \mathcal{G}^{(2,1)} & \mathcal{G}^{(2,2)} & \mathcal{G}^{(2,3)} \\ \mathcal{G}^{(3,1)} & \mathcal{G}^{(3,2)} & \mathcal{G}^{(3,3)} \end{pmatrix}, \tag{165}$$

in which $\mathcal{G}^{(1,1)}$, $\mathcal{G}^{(1,2)}$, $\mathcal{G}^{(2,1)}$ and $\mathcal{G}^{(2,2)}$ are of size $M \times M$, $\mathcal{G}^{(1,3)}$ and $\mathcal{G}^{(2,3)}$ are of size $M\times(N-2M)$, $\mathcal{G}^{(3,1)}$ and $\mathcal{G}^{(3,2)}$ are of size $(N-2M)\times M$, and $\mathcal{G}^{(3,3)}$ is of size $(N-2M)\times(N-2M)$.

The elements of $\mathcal{G}^{(1,1)}$, $\mathcal{G}^{(1,2)}$, $\mathcal{G}^{(2,1)}$ and $\mathcal{G}^{(2,2)}$ can be expressed as in (110)-(113), with only slight modifications due to the presence of the function $R$ (or equivalently $W_R$) in the definition of $\mathsf{F}_{i,j}$ (159):

$$\begin{aligned}\left[\mathcal{G}^{(1,1)}\right]_{i,j} &= \frac{Q_i(p_j)W_R(p_j)}{\theta(2p_j)}\sum_{\sigma=\pm}\frac{\sigma\,Q_i(p_j+\sigma\eta)\prod_{n=1}^6\theta(\sigma p_j+\frac{\eta}{2}-a_n)}{(-1)^N a(\sigma p_j)d(-\sigma p_j)} \\ &= \frac{-Q(p_j)W_R(p_j)\prod_{n=1}^6\theta(a_n)}{a(p_j)d(-p_j)a(-p_j)d(p_j)}\sum_{\sigma=\pm}\frac{\sigma\mathbf{A}_\varepsilon(\sigma p_j)Q(p_j-\sigma\eta)}{\theta(2p_j+\sigma\eta)\,\theta_s(p_j-\sigma\eta,q_i)\,\theta_s(p_j,q_i)}, \end{aligned}\tag{166}$$

$$\left[\mathcal{G}^{(1,2)}\right]_{i,j} = \delta_{i,j}\frac{Q_i(q_i)W_R(q_i)}{\theta(2q_i)}\sum_{\sigma=\pm}\frac{\sigma\,Q_i(q_i+\sigma\eta)\prod_{n=1}^6\theta(\sigma q_i+\frac{\eta}{2}-a_n)}{(-1)^N a(\sigma q_i)d(-\sigma q_i)}, \tag{167}$$

$$\begin{aligned}\left[\mathcal{G}^{(2,1)}\right]_{i,j} &= \frac{Q(p_j)W_R(p_j)}{\theta(2p_j)}\sum_{\sigma=\pm}\frac{(-1)^N\sigma\,Q(p_j+\sigma\eta)\prod_{n=1}^6\theta(\sigma p_j+\frac{\eta}{2}-a_n)}{\theta_s(p_j+\sigma\frac{\eta}{2},q_i+\frac{\eta}{2})\,\theta_s(p_j+\sigma\frac{\eta}{2},\gamma)a(\sigma p_j)d(-\sigma p_j)} \\ &\quad -\mathsf{X}_\gamma(q_i+\eta/2)\mathsf{M}_{\{a'\}}(\gamma,p_j), \end{aligned}\tag{168}$$

$$\begin{aligned}\left[\mathcal{G}^{(2,2)}\right]_{i,j} &= \delta_{i,j}\frac{Q_i(q_i+\eta)Q_i(q_i)W_R(q_i)}{(-1)^N a(q_i)d(-q_i)}\frac{\theta(\eta)\prod_{n=1}^6\theta(q_i+\frac{\eta}{2}-a_n)}{\theta_s(q_i+\frac{\eta}{2},\gamma)} \\ &\quad -\mathsf{X}_\gamma(q_i+\eta/2)\mathsf{M}_{\{a'\}}(\gamma,q_j), \end{aligned}\tag{169}$$

in which we have used the shortcut notation

$$\mathsf{X}_\gamma(\lambda) = \frac{Q(\gamma+\frac{\eta}{2})Q(\gamma-\frac{\eta}{2})R(\gamma)}{\theta_s(\gamma,\lambda)\,\theta(2\gamma)\prod_{n=1}^N\theta_s(\gamma,\xi_n)}. \tag{170}$$

Similarly, the elements of the blocks $\mathcal{G}^{(3,1)}$ and $\mathcal{G}^{(3,2)}$ are respectively

$$\begin{aligned}\left[\mathcal{G}^{(3,1)}\right]_{i,j} &= \sum_{k=1}^N \mathrm{Res}(\mathsf{F}_{i+2M,j};\xi_k) = -\sum_{\sigma=\pm}\mathrm{Res}(\mathsf{F}_{i+2M,j};p_j+\sigma\eta/2) - \mathrm{Res}(\mathsf{F}_{i+2M,j};\gamma) \\ &= \frac{Q(p_j)W_R(p_j)}{\theta(2p_j)}\sum_{\sigma=\pm}\frac{(-1)^N\sigma\,Q(p_j+\sigma\eta)\prod_{n=1}^6\theta(\sigma p_j+\frac{\eta}{2}-a_n)}{a(\sigma p_j)d(-\sigma p_j)\,\theta_s(p_j+\sigma\frac{\eta}{2},r_i)\,\theta_s(p_j+\sigma\frac{\eta}{2},\gamma)} \\ &\quad -\mathsf{X}_\gamma(r_i)\mathsf{M}_{\{a'\}}(\gamma,p_j), \end{aligned}\tag{171}$$

$$\left[\mathcal{G}^{(3,2)}\right]_{i,j} = \sum_{k=1}^N \mathrm{Res}(\mathsf{F}_{i+2M,j+M};\xi_k) = -\mathrm{Res}(\mathsf{F}_{i+2M,j+M};\gamma) = -\mathsf{X}_\gamma(r_i)\mathsf{M}_{\{a'\}}(\gamma,q_j). \tag{172}$$

The block $\mathcal{G}^{(1,3)}$ vanishes,

$$\left[\mathcal{G}^{(1,3)}\right]_{i,j} = \sum_{k=1}^{N} \text{Res}(\mathsf{F}_{i,j+2M}; \xi_k) = 0, \tag{173}$$

since the functions $\mathsf{F}_{i,j+2M}$, $1 \le i \le M$, $1 \le j \le N-2M$, have no other poles than at the points $\pm\xi_k$, $1 \le k \le N$. Finally, for the blocks $\mathcal{G}^{(2,3)}$ and $\mathcal{G}^{(3,3)}$, we obtain respectively

$$\left[\mathcal{G}^{(2,3)}\right]_{i,j} = \sum_{k=1}^{N} \text{Res}(\mathsf{F}_{i+M,j+2M}; \xi_k) = -\text{Res}(\mathsf{F}_{i+M,j+2M}; \gamma)$$
$$= -\mathsf{X}_\gamma(q_i + \eta/2)\,\mathsf{M}_{\{a'\}}(\gamma, w_{2M+j}), \tag{174}$$

$$\left[\mathcal{G}^{(3,3)}\right]_{i,j} = \sum_{k=1}^{N} \text{Res}(\mathsf{F}_{i+2M,j+2M}; \xi_k)$$
$$= -(\delta_{i,j} + \delta_{i,j+1})\,\text{Res}(\mathsf{F}_{i+2M,j+2M}; r_i) - \text{Res}(\mathsf{F}_{i+2M,j+2M}; \gamma)$$
$$= \frac{Q(r_i + \frac{\eta}{2})Q(r_i - \frac{\eta}{2})R_i(r_i)}{\prod_{n=1}^{N} \theta_s(r_i, \xi_n)\,\theta(r_i, \gamma)\,\theta(2w_{2M+j})}$$
$$\times \left[\delta_{i,j+1}\prod_{n=1}^{6}\theta(r_i - a_n') - \delta_{i,j}\prod_{n=1}^{6}\theta(r_i + a_n')\right] - \mathsf{X}_\gamma(r_i)\,\mathsf{M}_{\{a'\}}(\gamma, w_{2M+j}), \tag{175}$$

in which we have defined $R_i(\lambda) = R(\lambda)/\theta_s(\lambda, r_i)$.

Once again, when we particularise $\{q_1, \ldots, q_M\}$ to be a solution of the Bethe equations (92), the expression (167) vanishes, so that $\mathcal{G}^{(1,2)} = 0$, and therefore, taking also into account (173),

$$\det_N[\mathcal{X} \cdot \mathcal{M}] = \det_M\left[\mathcal{G}^{(1,1)}\right] \cdot \det_{M+N}\left[\bar{\mathcal{G}}^{(2,2)}\right], \tag{176}$$

in which

$$\bar{\mathcal{G}}^{(2,2)} = \begin{pmatrix} \mathcal{G}^{(2,2)} & \mathcal{G}^{(2,3)} \\ \mathcal{G}^{(3,2)} & \mathcal{G}^{(3,3)} \end{pmatrix} \tag{177}$$

is a $(M+N) \times (M+N)$ matrix which does not depend on $P$. The remaining part of the computation, which consists in considering the ratio of $_\varepsilon\langle P | Q \rangle_\varepsilon$ by $_\varepsilon\langle \widetilde{P} | Q \rangle_\varepsilon$, in which $\widetilde{P}$ is another function of the form (82) with roots $\tilde{p}_1, \ldots, \tilde{p}_M$, is then similar as for the other cases.

## 5.4 Proof of theorem 4.2

Let now $\varepsilon' = -\varepsilon$. In that case,

$$\frac{\mathsf{A}_\epsilon(\frac{\eta}{2} + \xi_n)}{\mathsf{A}_{\epsilon'}(\frac{\eta}{2} - \xi_n)} = \frac{\mathsf{A}_\epsilon(\frac{\eta}{2} + \xi_n)}{\mathsf{A}_{-\epsilon}(\frac{\eta}{2} - \xi_n)} = 1, \tag{178}$$

for any choice of the boundary parameters, so that the representation (78) is independent of the boundary parameters:

$$_\varepsilon\langle P | Q \rangle_{-\varepsilon} = \prod_{n=1}^{N}(PQ)(\xi_n^{(0)})\,\frac{\det_{1\le i,j\le N}\left[\Theta_N^{(j)}(\xi_i^{(1)}) - \frac{P(\xi_i^{(1)})Q(\xi_i^{(1)})}{P(\xi_i^{(0)})Q(\xi_i^{(0)})}\Theta_N^{(j)}(\xi_i^{(0)})\right]}{\det_{1\le i,j\le N}\left[\Theta_N^{(j)}(\xi_i)\right]}. \tag{179}$$

For $M' + M = N - 1$, let us now choose a basis of $\Xi_N$ as in (81), with

$$W(\lambda) = P(\lambda)Q(\lambda)\,\theta_s(\lambda, w_N), \tag{180}$$

for any arbitrary $w_N$. We get

$$\,_{\varepsilon}\langle P \,|\, Q \rangle_{-\varepsilon} = \prod_{n=1}^{N} \frac{(PQ)(\xi_n^{(0)})(PQ)(\xi_n^{(1)})}{(PQ)(\xi_n)} \frac{\det_{1 \leq i,j \leq N} \left[ \frac{\theta_s(\xi_i^{(1)}, w_N)}{\theta_s(\xi_i^{(1)}, w_j)} - \frac{\theta_s(\xi_i^{(0)}, w_N)}{\theta_s(\xi_i^{(0)}, w_j)} \right]}{\det_{1 \leq i,j \leq N} \left[ \frac{\theta_s(\xi_i, w_N)}{\theta_s(\xi_i, w_j)} \right]} . \tag{181}$$

The last column in the determinant of the numerator of (181) being identically 0, we obtain the result.

## 6 Conclusion

We have here solved the open XYZ spin 1/2 chain by the new Separation of Variables approach of [1, 2], and computed in this framework the scalar products of separate states, a class of states which contains as a sub-class the eigenstates of the model. As usual for models solved by SoV, these scalar products of separate states have a simple determinant representation, see Proposition 4.1. However, as usual, this determinant depends in an intricate way on the inhomogeneity parameters of the model (the latter label its rows, see (78)), which is not very convenient for the consideration of the homogeneous and thermodynamic limits. Our aim was therefore to transform the representation (78) into a more convenient one for physical applications such as the computation of correlation functions.

We have considered in particular the case in which the 6 boundary parameters parametrising the two boundary fields satisfy one constraint (65), so that the transfer matrix spectrum and eigenstates can be partly described in terms of a usual $TQ$-equation (67), with a $Q$-function being an even elliptic polynomial of the form (66). Focusing on separate states constructed from even elliptic polynomials of the form (66) under the constraint (65), we have shown how to transform the representation (78) into a new one (83), in the case in which one of the separate states is an eigenstate of the model. The representation (83), which constitutes the main result of our paper, is a direct elliptic generalisation of the determinant representations which have been previously obtained in the rational [4] and trigonometric [5] cases, i.e. for the open XXX and XXZ chain respectively. These representations are generalisations of the famous Slavnov determinant representation [6], and were directly used in the open XXX and XXZ cases for the computation of correlation functions [90–92].

The approach we used here is however not the direct elliptic generalisation of the approach used in [4, 5]: the problem comes from the fact that the initial scalar product representation (78) depends on more boundary parameters in the present XYZ case than in the XXX or XXZ cases [4, 5], which makes it difficult to use symmetry arguments as in [4, 5]. We instead proposed a new method allowing to almost directly transform (78) into (83): we multiplied the matrix appearing in the initial representation (95) issued from (78) by a well-chosen matrix $\mathcal{X}$, so that the resulting product can be computed via the residue theorem on certain odd elliptic functions (105), and simplifies when the Bethe equations are satisfied for one the two sets of parameters labelling the separate states.

The new scalar product formulas that we have obtained here constitute the first step towards the computation of the correlation functions. Building on this result, we have implemented the generalisation of the results of [90–92] to the XYZ open spin chain in [106].

# A  The trigonometric limit

We consider here the trigonometric limit $\Im\omega \to +\infty$, in which, if $\lambda$ remains finite,

$$\lim_{\Im\omega\to+\infty} \frac{e^{-i\pi\omega/4}}{2}\theta_1(\lambda|\omega) = \sin\lambda\,, \qquad \lim_{\Im\omega\to+\infty} \frac{e^{-i\pi\omega/4}}{2}\theta_2(\lambda|\omega) = \cos\lambda\,, \qquad (A.1)$$

$$\lim_{\Im\omega\to+\infty} \theta_3(\lambda|\omega) = \lim_{\Im\omega\to+\infty} \theta_4(\lambda|\omega) = 1\,. \qquad (A.2)$$

Let us define new boundary parameters $\varphi_\pm, \psi_\pm$ and $\tau_\pm$ by

$$\alpha_1^\pm = -i\varphi_\mp\,, \qquad \alpha_2^\pm = i\psi_\mp - \epsilon\frac{\pi}{2}\,, \qquad \alpha_3^\pm = i\tau_\mp + \epsilon\frac{\pi}{2} + \frac{\pi\omega}{2}\,, \qquad (A.3)$$

for some arbitrary sign $\epsilon$, and we suppose that these new boundary parameters $\varphi_\pm, \psi_\pm, \tau_\pm$ remain finite in the limit $\Im\omega \to +\infty$. Then, the boundary parameters $c_\pm^{x,y,z}$ (9) behave in this limit in terms of (A.3) as

$$\lim_{\Im\omega\to+\infty} 2e^{i\pi\omega/4}c_\mp^x = \frac{-i\cosh(\tau^\pm)}{\sinh(\varphi^\pm)\cosh(\psi^\pm)}\,, \qquad (A.4)$$

$$\lim_{\Im\omega\to+\infty} 2e^{i\pi\omega/4}c_\mp^y = \frac{-i\sinh(\tau^\pm)}{\sinh(\varphi^\pm)\cosh(\psi^\pm)}\,, \qquad (A.5)$$

$$\lim_{\Im\omega\to+\infty} c_\mp^z = -i\frac{\cosh(\varphi^\pm)\sinh(\psi^\pm)}{\sinh(\varphi^\pm)\cosh(\psi^\pm)}\,. \qquad (A.6)$$

Hence, in the limit $\Im\omega \to +\infty$, with $\eta, \varphi_\pm, \psi_\pm, \tau_\pm$ remaining finite, the Hamiltonian (27) degenerates into the Hamiltonian of the open XXZ spin chain:

$$H_{\text{XXZ}} = \sum_{n=1}^{N-1}\left[\sigma_n^x\sigma_{n+1}^x + \sigma_n^y\sigma_{n+1}^y + \Delta\,\sigma_n^z\sigma_{n+1}^z\right] + \sum_{a\in\{x,y,z\}}\left[\tilde{h}_-^a\,\sigma_1^a + \tilde{h}_+^a\,\sigma_N^a\right], \qquad (A.7)$$

in which

$$\Delta = \cos\eta = \cosh\tilde\eta\,, \qquad \text{with} \quad \eta = i\tilde\eta\,, \qquad (A.8)$$

$$\tilde{h}_\pm^x = \sinh\tilde\eta\,\frac{\cosh\tau_\pm}{\sinh\varphi_\pm\,\cosh\psi_\pm}\,, \qquad (A.9)$$

$$\tilde{h}_\pm^y = i\sinh\tilde\eta\,\frac{\sinh\tau_\pm}{\sinh\varphi_\pm\,\cosh\psi_\pm}\,, \qquad (A.10)$$

$$\tilde{h}_\pm^z = \sinh\tilde\eta\,\coth\varphi_\pm\,\tanh\psi_\pm\,. \qquad (A.11)$$

Moreover, still in this limit, if in addition $\lambda = iu$ remains finite, the R-matrix (1) degenerates, up to a global normalisation, into the 6-vertex R-matrix,

$$R_{\text{XXZ}}(u) = \begin{pmatrix} \sinh(u+\tilde\eta) & 0 & 0 & 0 \\ 0 & \sinh u & \sinh\tilde\eta & 0 \\ 0 & \sinh\tilde\eta & \sinh u & 0 \\ 0 & 0 & 0 & \sinh(u+\tilde\eta) \end{pmatrix}\,, \qquad (A.12)$$

and the boundary K-matrices (13) become

$$K_\pm(\lambda\mp\eta/2) \xrightarrow[\Im\omega\to+\infty]{} K_{\text{XXZ}}(-i\lambda;\varphi_\mp,\psi_\mp,\tau_\mp) \qquad (A.13)$$

in which

$$K_{\text{XXZ}}(u;\varphi,\psi,\tau) = \cosh u\,\mathbb{I} + \frac{\cosh\varphi\,\sinh\psi}{\sinh\varphi\,\cosh\psi}\,\sinh u\,\sigma^z$$
$$+ \frac{\sinh(2u)}{2\sinh\varphi\,\cosh\psi}\left[\cosh\tau\,\sigma^x + i\sinh\tau\,\sigma^y\right] \tag{A.14}$$
$$= \begin{pmatrix} \cosh u + \sinh u\frac{\cosh\varphi\,\sinh\psi}{\sinh\varphi\,\cosh\psi} & e^\tau\frac{\sinh 2u}{2\sinh\varphi\,\cosh\psi} \\ e^{-\tau}\frac{\sinh 2u}{2\sinh\varphi\,\cosh\psi} & \cosh u - \sinh u\frac{\cosh\varphi\,\sinh\psi}{\sinh\varphi\,\cosh\psi} \end{pmatrix}$$

is the K-matrix of the open XXZ spin chain. Finally, the coefficient (62) becomes

$$A_\varepsilon(\lambda) \xrightarrow[\Im\omega\to+\infty]{} e^{-(\epsilon_3^+ + \epsilon_3^-)(u - \frac{\tilde\eta}{2})}\prod_{\sigma=\pm}\frac{\sinh(u - \frac{\tilde\eta}{2} + \epsilon_1^\sigma\varphi_\sigma)\cosh(u - \frac{\tilde\eta}{2} - \epsilon_2^\sigma\psi_\sigma)}{\sinh(\epsilon_1^\sigma\varphi_\sigma)\cosh(-\epsilon_2^\sigma\psi_\sigma)}\,. \tag{A.15}$$

For the particular choice $\epsilon_3^+ = -\epsilon_3^-$, $\epsilon_1^\pm = \epsilon_{\varphi^\mp}$, $\epsilon_2^\pm = \pm\epsilon_{\psi^\mp}$, we recover the convention of our previous XXZ papers [5,92]:[10] the constraint (65) becomes

$$\epsilon_3^+(\tau_+ - \tau_-) + \sum_{\sigma=\pm}\left[\epsilon_{\varphi_\sigma}\varphi_\sigma + \sigma\epsilon_{\psi_\sigma}(\psi_\sigma + i\epsilon\frac{\pi}{2})\right] + (N - 2M - 1)\tilde\eta = 0\,, \tag{A.16}$$

with $\epsilon_{\varphi_+}\epsilon_{\varphi_-}\epsilon_{\psi_+}\epsilon_{\psi_-} = 1$; under this constraint, we recover the XXZ analog of Proposition 3.2 for trigonometric polynomials $Q$ of the form

$$Q(u) = \prod_{n=1}^M \sinh(u - \tilde q_n)\sinh(u + \tilde q_n)\,, \qquad \tilde{\mathbf{q}} = (\tilde q_1,\ldots,\tilde q_M) \in \mathbb{C}^M\,. \tag{A.17}$$

It is now easy to see that Theorem 4.1 reduces to Theorem 5.2 of [5] in the trigonometric limit.[11]

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
