# Peer review of "The open XYZ spin 1/2 chain: Separation of Variables and scalar products for boundary fields related by a constraint"

_SciPost Physics, doi:SciPost Phys. 19, 090 (2025)_

## Round 2 · Referee Report · Anonymous (Referee 1) · 2025-5-6

Strengths

1 - The paper proposes a novel method to generalize the Separation of Variables (SoV) approach from the $XXX$ and $XXZ$ spin chains to the open $XYZ$ spin chain with the elliptic R-matrix. 2- The authors successfully derive scalar products of separate states for the open $XYZ$ spin chain, achieving an elegant result (4.14).

Weaknesses

The primary weakness is the lack of consistency verification between the new results and prior work on the $XXX/XXZ$ cases.

Report

The authors present an extension of the SoV framework, generalizing it from $XXX/XXZ$ spin chains with rational/trigonometric R-matrices to the open $XYZ$ spin chain with the elliptic R-matrix. They develop a novel method to compute scalar products of separate states using the residue theorem and obtain the elegant form of the result (4.14). The paper merits publication in SciPost Physics after addressing the following revisions:
1 - Consistency checks with degenerate cases. To strengthen the paper’s impact, the authors should explicitly verify that their results degenerate to established results on $XXX$ or $XXZ$ spin chains under appropriate limits.
2 - In the last paragraph on page 7, the phrase "...for almost any choice of the co-vector..." is ambiguous. The authors should specify the meaning of "almost any".
3 - For Eq. (3.27), there is a typo in "{1,...,n}".
4 - In the last paragraph of Section 3 on page 10, the phrase "...in terms of Q-functions solution..." seems a grammar error.
5 - In the last paragraph of Section 3 on page 10, the statement "... at least for some range of the boundary parameters" should be expanded to clarify whether completeness fails for specific parameter ranges. A brief discussion would resolve this ambiguity.

Requested changes

Please see the Report part.

Recommendation

Ask for major revision

  • validity: high
  • significance: high
  • originality: top
  • clarity: top
  • formatting: perfect
  • grammar: perfect

Author:  Giuliano Niccoli  on 2025-07-30  [id 5692]

(in reply to Report 1 on 2025-05-06)

Dear Editor,
We would like to thanks both the referees for their attentive reading and analysis of our manuscript which has allowed us to improve it, answering to their requests of clarifications, verifications and typos. We have taken into account mainly all the requirements of the two referees and we detail them in the following
Modifications done to take into account the first referee’s report:
1- We have added an appendix to address the trigonometric limit on the elliptic case and we have verified that in this limit the results and formulae derived in the XYZ case reproduces those of the XXZ case. The limit from the XXZ case to the XXX is then easy and it works as well.
2- We have added a footnote at page 7 to clarify the meaning of “almost any”. The main point is that the linear independence of the $2^N$ covectors defined in (3.17), is proven by showing that the determinant of the matrix of coefficients of these covectors is nonzero. This determinant is a polynomial in the inhomogeneity parameters and in the coordinates of the reference convector. So, we just need to prove that it is nonzero for some special values of these parameters to have that it is not identically zero. So that the “almost any” means that these covectors are independent and so form a basis for any values of these parameters with the exceptions of the zeros of this multivariable polynomial, which is a codimension 1 hypersurface in the full space of inhomogeneities and reference convector coordinates.
3- We have corrected the typo evidenced by the referee.
4- We have corrected the typo evidenced by the referee.
5- One should point out that the analysis developed in [93,94] to advance their conjecture is mainly a numerical analysis on small chains. While this study is developed without further restrictions on the boundary parameters a part (3.35), the absence so far in the literature of an analytic study does not allow to answer if completeness fails for specific parameter ranges. So, we have changed our last sentence and removed the statement "at least for some range of the boundary parameters" and stated explicitly that the conjecture is based only on numerical analysis.

---

## Round 2 · Referee Report · Anonymous (Referee 2) · 2025-6-6

Strengths

1-Application of the SoV method to a new integrable model.

2-Derivation of a Slavnov-type determinant formula for scalar products in the open XYZ spin chain.

Weaknesses

1-Section 3 heavily relies on prior derivations for the XXX and XXZ models, making it difficult to follow without familiarity with those results.
2-The Slavnov determinant representation is not applicable across the full spectrum of a given model.

Report

The authors investigate the open XYZ spin chain using the Separation of Variables (SoV) method. They begin by diagonalizing the boundary transfer matrix via SoV, which serves as a natural generalization of earlier results for the open XXX and XXZ models. The eigenvalues, expressed as elliptic polynomials in the spectral parameter, are obtained by solving an inhomogeneous system of quadratic equations. In certain cases, solutions can also be derived from homogeneous TQ-equations, which exist only if the six boundary parameters satisfy a specific condition dependent on the number of Bethe roots.

In the SoV framework, scalar products of separable states naturally take a determinant form, where the matrix rows are labeled by the inhomogeneity parameters. However, this representation complicates the homogeneous limit, limiting its practical applicability. The main contribution of the paper is the derivation of Slavnov-type determinant expressions for scalar products involving one on-shell vector. In this formulation, the matrix elements are labeled not by inhomogeneities but by more physical Bethe roots, which significantly enhances its utility in applications.

Requested changes

Comments and Suggestions:

1, In equations (3.7)–(3.10), the notation det_q appears inconsistent with earlier usage.

2, In the proof of Proposition 3.2, it would be helpful to clearly indicate where condition (3.35) should be applied.

3, In equation (4.5), the left-hand side is independent of n, while the right-hand side depends on it—possibly a product symbol is missing.

Recommendation

Publish (easily meets expectations and criteria for this Journal; among top 50%)

  • validity: high
  • significance: good
  • originality: high
  • clarity: high
  • formatting: excellent
  • grammar: -

Author:  Giuliano Niccoli  on 2025-07-30  [id 5693]

(in reply to Report 2 on 2025-06-06)

Dear Editor,
We would like to thanks both the referees for their attentive reading and analysis of our manuscript which has allowed us to improve it, answering to their requests of clarifications, verifications and typos. We have taken into account mainly all the requirements of the two referees and we detail them in the following
Modifications done to take into account the second referee’s report:

1- In eq (3.7)-(3.10), $det_q$ stays for quantum determinant, it is central, i.e. just a number and it was defined in eq (2.22) and it is the product of the coefficients $a(\lambda)$ and $d(\lambda)$ defined in (2.23).
2- We have clarified that the condition (3.35), in Proposition 3.2, is required to have that the entire function $\tau(\lambda)$ is indeed an elliptic polynomial of degree 2N+6 once it satisfied the TQ functional equation (3.37), which is one of the requirements that it has to satisfy to be a transfer matrix eigenvalue.
3- We have corrected the misprint remarked by the referee in equation (4.5).

---

## Round 3 · Author Response

We would like to thanks both the referees for their attentive reading and analysis of our manuscript which has allowed us to improve it, answering to their requests of clarifications, verifications and typos. We have taken into account mainly all the requirements of the two referees and we detail them in the following

---

## Round 3 · List of Changes

1- We have added an appendix to address the trigonometric limit on the elliptic case and we have verified that in this limit the results and formulae derived in the XYZ case reproduces those of the XXZ case. The limit from the XXZ case to the XXX is then easy and it works as well.
2- We have added a footnote at page 7 to clarify the meaning of “almost any”. The main point is that the linear independence of the $2^N$ covectors defined in (3.17), is proven by showing that the determinant of the matrix of coefficients of these covectors is nonzero. This determinant is a polynomial in the inhomogeneity parameters and in the coordinates of the reference convector. So, we just need to prove that it is nonzero for some special values of these parameters to have that it is not identically zero. So that the “almost any” means that these covectors are independent and so form a basis for any values of these parameters with the exceptions of the zeros of this multivariable polynomial, which is a codimension 1 hypersurface in the full space of inhomogeneities and reference convector coordinates.
3- We have corrected the typo evidenced by the referee.
4- We have corrected the typo evidenced by the referee.
5- One should point out that the analysis developed in [93,94] to advance their conjecture is mainly a numerical analysis on small chains. While this study is developed without further restrictions on the boundary parameters a part (3.35), the absence so far in the literature of an analytic study does not allow to answer if completeness fails for specific parameter ranges. So, we have changed our last sentence and removed the statement "at least for some range of the boundary parameters" and stated explicitly that the conjecture is based only on numerical analysis.
Modifications done to take into account the second referee’s report:
1- In eq (3.7)-(3.10), $det_q$ stays for quantum determinant, it is central, i.e. just a number and it was defined in eq (2.22) and it is the product of the coefficients $a(\lambda)$ and $d(\lambda)$ defined in (2.23).
2- We have clarified that the condition (3.35), in Proposition 3.2, is required to have that the entire function $\tau(\lambda)$ is indeed an elliptic polynomial of degree 2N+6 once it satisfied the TQ functional equation (3.37), which is one of the requirements that it has to satisfy to be a transfer matrix eigenvalue.
3- We have corrected the misprint remarked by the referee in equation (4.5).

---

## Editorial Decision

published